# Identification of TMEM206 proteins as pore of PAORAC/ASOR acid-sensitive chloride channels

**Florian Ullrich[1,2], Sandy Blin[1,2], Katina Lazarow[1], Tony Daubitz[1,2], Jens Peter von Kries[1], Thomas J Jentsch[1,2,3]\***

[1]Leibniz-Forschungsinstitut für Molekulare Pharmakologie (FMP), Berlin, Germany;
[2]Max-Delbrück-Centrum für Molekulare Medizin (MDC), Berlin, Germany;
[3]NeuroCure Cluster of Excellence, Charité Universitätsmedizin, Berlin, Germany

**Abstract** Acid-sensing ion channels have important functions in physiology and pathology, but the molecular composition of acid-activated chloride channels had remained unclear. We now used a genome-wide siRNA screen to molecularly identify the widely expressed acid-sensitive outwardly-rectifying anion channel PAORAC/ASOR. ASOR is formed by TMEM206 proteins which display two transmembrane domains (TMs) and are expressed at the plasma membrane. Ion permeation-changing mutations along the length of TM2 and at the end of TM1 suggest that these segments line ASOR's pore. While not belonging to a gene family, TMEM206 has orthologs in probably all vertebrates. Currents from evolutionarily distant orthologs share activation by protons, a feature essential for ASOR's role in acid-induced cell death. TMEM206 defines a novel class of ion channels. Its identification will help to understand its physiological roles and the diverse ways by which anion-selective pores can be formed.
DOI: https://doi.org/10.7554/eLife.49187.001

**\*For correspondence:**
jentsch@fmp-berlin.de

**Competing interests:** The authors declare that no competing interests exist.

## Introduction

Chloride is by far the most abundant anion in animals. Its concentration can vary substantially between the extracellular space, the cytoplasm and various intracellular organelles, resulting in concentration gradients across membranes separating these compartments. Negatively charged chloride can cross biological membranes only with the help of membrane-spanning proteins such as Cl⁻ channels, which allow passive diffusion of Cl⁻ along its electrochemical gradient, or transporter proteins that couple the movement of Cl⁻ to that of other ions and can thereby establish electrochemical gradients.

Chloride channels fulfill a broad range of biological functions, including the homeostasis of cell volume, vesicular acidification, transepithelial transport and cellular signaling (*Jentsch and Pusch, 2018*; *Jentsch et al., 2002*). Elucidation of these roles has been greatly facilitated by the molecular identification of the underlying channel proteins, a discovery process that began in the late 1980's and is still ongoing. Chloride channels are molecularly and structurally very diverse and lack a defining 'signature' pattern. More than six unrelated Cl⁻ channel families are known, prominently including voltage-regulated CLC channels (*Jentsch and Pusch, 2018*; *Jentsch et al., 1990*), bestrophin (*Hartzell et al., 2008*; *Sun et al., 2002*) and TMEM16 (*Caputo et al., 2008*; *Schroeder et al., 2008*; *Yang et al., 2008*) Ca²⁺-activated Cl⁻ channels, as well as LRRC8/VRAC volume-regulated anion channels (*Qiu et al., 2014*; *Voss et al., 2014*). However, several Cl⁻ currents biophysically characterized in mammalian cells still lack molecular correlates, severely hindering the elucidation of their cellular and organismal functions.

We set out to identify the protein(s) mediating a widely expressed, strongly outwardly-rectifying plasma membrane $Cl^-$ current $I_{Cl,H}$ that is detectable only upon marked extracellular acidification and which may play a role in acid-induced cell death (*Auzanneau et al., 2003*; *Capurro et al., 2015*; *Lambert and Oberwinkler, 2005*; *Sato-Numata et al., 2013*; *Wang et al., 2007*). The underlying channel is most frequently called ASOR (for *A*cid-*S*ensitive *O*utwardly *R*ectifying anion channel; *Wang et al., 2007*), although the earlier name PAORAC (*P*roton-*A*ctivated *O*utwardly *R*ectifying *A*nion *C*hannel; *Lambert and Oberwinkler, 2005*; *Ma et al., 2008*) provides a better description. At room temperature, ASOR currents are only observable when external pH ($pH_o$) drops below 5.5, but the threshold of activation shifts to ~6.0 at 37°C (*Sato-Numata et al., 2013*). Analysis of the steep pH-dependence suggests that 3–4 protons are required to open the channel (*Capurro et al., 2015*; *Lambert and Oberwinkler, 2005*). ASOR currents are strongly outwardly rectifying with almost no currents being observable at negative-inside voltages. This rectification results both from voltage-dependent gating that operates on a 100 ms time scale and from an outwardly-rectifying single channel conductance (*Lambert and Oberwinkler, 2005*) (although others described that single ASOR channels are not rectifying, for example *Wang et al., 2007*). ASOR currents ($I_{Cl,H}$) display an $SCN^->I^->NO_3^->Br^->Cl^-$ permeability sequence. The channel can be blocked by various $Cl^-$ transport inhibitors such as DIDS (4,4'-diisothiocyano-2,2'-stilbenedisulfonic acid) and niflumic acid (*Capurro et al., 2015*; *Lambert and Oberwinkler, 2005*) and by other compounds such as phloretin (*Wang et al., 2007*) and pregnenolone sulfate (PS) (*Drews et al., 2014*). However, none of these inhibitors is specific for ASOR.

The observation of $I_{Cl,H}$ currents in every investigated mammalian cell type (*Auzanneau et al., 2003*; *Capurro et al., 2015*; *Fu et al., 2013*; *Lambert and Oberwinkler, 2005*; *Ma et al., 2008*; *Nobles et al., 2004*; *Sato-Numata et al., 2013*; *Valinsky et al., 2017*; *Wang et al., 2007*; *Yamamoto and Ehara, 2006*) suggests that ASOR may be expressed in all tissues. Such a wide expression pattern indicates that this channel has important physiological functions. However, only few mammalian cells are physiologically exposed to an extracellular pH that is acidic enough to open ASOR. It was therefore proposed (*Wang et al., 2007*) that the channel rather plays a role in pathologies such as cancer or ischemic stroke in which $pH_o$ can drop to pH 6.5 or below (*Gillies et al., 2018*; *Kato et al., 2013*; *Nedergaard et al., 1991*; *Thews and Riemann, 2019*; *Xiong et al., 2004*). Indeed, ASOR inhibitors blunted cell swelling and cell death provoked by pro-longed exposure to acidic $pH_o$ (*Sato-Numata et al., 2014*; *Wang et al., 2007*), suggesting that ASOR-mediated chloride influx may worsen the outcome of ischemic stroke. It seems, however, counterintuitive that a channel that enhances cell death under pathological conditions confers an evolutionary advantage. Alternatively, ASOR might not only be present at the plasma membrane, but also in intracellular compartments such as lysosomes where it would be exposed to an appropriate acidic pH (*Lambert and Oberwinkler, 2005*).

The molecular identity of ASOR had remained unknown, with several candidates such as LRRC8/VRAC anion channels or ClC-3 $Cl^-/H^+$ exchangers having been excluded previously (*Lambert and Oberwinkler, 2005*; *Sato-Numata et al., 2016*; *Sato-Numata et al., 2013*). Using a genome-wide siRNA screen we now identified TMEM206 as essential ASOR component. TMEM206, which has two transmembrane domains, was necessary and sufficient for the formation of functional ASOR channels. All tested TMEM206 orthologs from other vertebrates form acid-activated channels which, however, differ moderately in their biophysical properties. Ion-selectivity changing mutations suggest that TMEM206's transmembrane domains, in particular TM2, form ASOR's pore. Finally, partial protection of *TMEM206*$^{-/-}$ cells from acid-induced cell death indicates that ASOR/TMEM206 channels play a detrimental role in pathological conditions that are associated with tissue acidosis.

## Results

### Identification of TMEM206 as crucial ASOR component

To identify the protein(s) constituting ASOR, we performed a genome-wide siRNA screen using an optical assay for acute ASOR-mediated iodide influx into HeLa cells (*Figure 1A*). Intracellular iodide was detected by fluorescence quenching of inducibly expressed, halide-sensitive and relatively pH-insensitive $E^2GFP$ (*Arosio et al., 2007*). We activated ASOR by exposing cells to acidic extracellular pH (5.0) and simultaneously depolarized their plasma membrane to maximize ASOR currents. To this

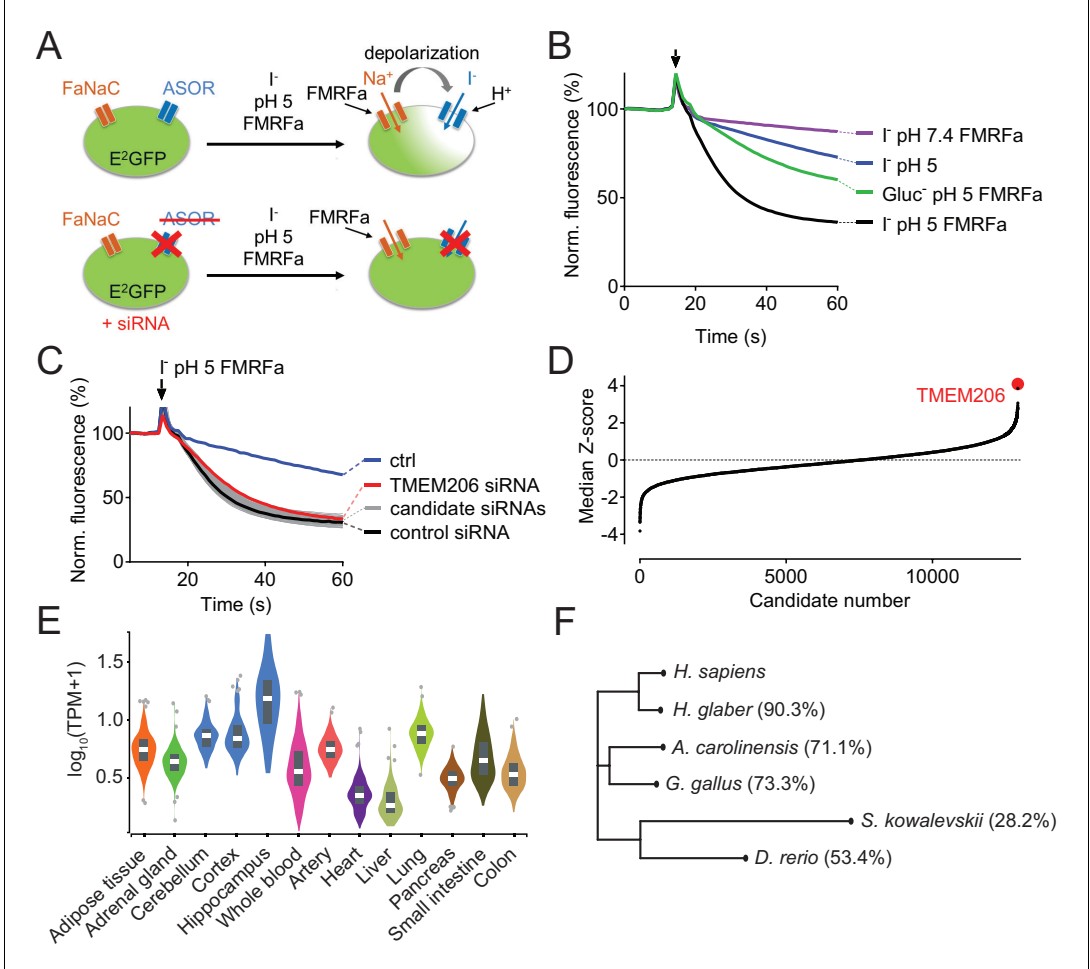

**Figure 1.** Identification of TMEM206 as ASOR component. (**A**) Screening assay. Engineered HeLa cells inducibly expressing iodide-sensitive $E^2GFP$ and FRMFamide-gated $Na^+$ channel FaNaC were acutely exposed to an acidic solution (pH 5) containing 20 µM FMRFamide and 100 mM $I^-$. Iodide influx through ASOR is stimulated both by acidic pH and the depolarization caused by FaNaC-mediated $Na^+$-influx and induces quenching of $E^2GFP$ fluorescence. Fluorescence quenching is reduced by ASOR knock-down. (**B**) Assay verification under the conditions used for screening. Addition of $I^-$ and FRMFamide at pH 5 (arrow) induces rapid quenching of fluorescence. Less rapid quenching upon omission of either $I^-$, FMRFamide or acidic pH suggests that it is caused by $I^-$ influx through ASOR, as further supported by inhibition by PS (pregnenolone sulfate) and DIDS (4,4'-diisothiocyano-2,2'-stilbenedisulfonic acid) (*Figure 1—figure supplement 1*). (**C**) Fluorescence curves from a 384-well plate treated with siRNA against 280 genes, including TMEM206. Fluorescence quenching is specifically slowed by siRNA against TMEM206. (**D**) Distribution of median Z-scores (mean of 3 replicates) from filtered hits (see Materials and methods). TMEM206 was the top hit. (**E**) Tissue expression of TMEM206 extracted from the GTEx database (https://gtexportal.org/home/; TPM, transcripts per million). (**F**) Dendrogram depicting similarity between TMEM206 orthologs from human (*Homo sapiens*), African naked mole-rat (*Heterocephalus glaber*), chicken (*Gallus gallus*), green anole lizard (*Anolis carolinensis*), zebrafish (*Danio rerio*) and from the hemichordate acorn worm (*Saccoglossus kowalewski*). Amino-acid sequence identity to human TMEM206 given in brackets. Dendrogram based on a Clustal Omega protein alignment fed into the Simple Phylogeny tool at EMBL-EBI (https://www.ebi.ac.uk/services).

DOI: https://doi.org/10.7554/eLife.49187.002

The following source data and figure supplement are available for figure 1:

**Source data 1.** Raw data for *Figure 1B-E*.
DOI: https://doi.org/10.7554/eLife.49187.004

**Figure supplement 1.** Verification of the assay for the genome-wide siRNA screen.
DOI: https://doi.org/10.7554/eLife.49187.003

end, we used an approach which we had used previously (*Scheel et al., 2005*) to activate ClC-5, a $Cl^-/H^+$-exchanger that displays a similarly strong outward rectification. In HeLa cells we stably co-expressed $E^2GFP$ with FaNaC (*Lingueglia et al., 1995*), a ligand-gated $Na^+$ channel from the snail *Cornu aspersum* that is closed until it is activated by the cognate neurotransmitter peptide Phe-Met-

Arg-Phe-NH$_2$ (FMRFamide). FMRFamide application leads to Na$^+$ influx that acutely depolarizes the membrane to inside-positive voltages. Concomitant application of 100 mM iodide, pH 5.0 and 20 µM FMRFamide to our engineered HeLa cells induced fast quenching of E$^2$GFP fluorescence (*Figure 1B*). Control experiments omitting either iodide, acidic pH$_o$, or FMRFamide indicated that a major component of quenching is owed to iodide influx through an acidic pH- and depolarization-dependent process such as ASOR (*Figure 1B*, *Figure 1—figure supplement 1A–D*). This notion was further supported by applying the inhibitors pregnenolone sulfate and DIDS (*Drews et al., 2014*) (*Figure 1—figure supplement 1E,F*).

Having established a sensitive assay for ASOR function, we performed a genome-wide siRNA screen using a commercial library containing pools of four siRNAs per gene (*Figure 1C*). The screen was carried out in triplicate and results were ranked according to the maximal slope of quenching. The top hit was TMEM206, a so far uncharacterized membrane protein (*Figure 1D*). The wide tissue expression pattern of TMEM206 indicated by public databases (*Figure 1E*) agreed with the finding that $I_{Cl,H}$ has been found in every cell studied so far (*Auzanneau et al., 2003*; *Capurro et al., 2015*; *Fu et al., 2013*; *Lambert and Oberwinkler, 2005*; *Ma et al., 2008*; *Nobles et al., 2004*; *Sato-Numata et al., 2013*; *Valinsky et al., 2017*; *Wang et al., 2007*). TMEM206 has orthologs in vertebrates and in the hemichordate *Saccoglossus kowalevskii* (acorn worm) (*Figure 1F*), but there are no homologs in simpler animals or other kingdoms of life. In contrast to many other ion channels, TMEM206 does not form a gene family because it lacks paralogs within the same species (i.e. it is a 'single gene').

After transfection of GFP-tagged human TMEM206 (hTMEM206) into HeLa cells a significant fraction of the protein was found at the plasma membrane (*Figure 2A*). Additionally, TMEM206 appeared to be expressed in intracellular membranes, which, however, might result from heterologous overexpression. Hydropathy analysis suggested the presence of two transmembrane spans (*Figure 2B*). To determine the transmembrane topology of TMEM206, we tested the accessibility of added epitopes by immunofluorescence. Detection of both N- or C-terminally added GFP required permeabilization of the plasma membrane (*Figure 2C,D*), suggesting that both the amino- and the carboxy-terminus of TMEM206 face the cytosol.

A cytosolic localization of the N-terminus is further supported by public databases showing that the amino-terminus of human and rodent TMEM206 can be phosphorylated (*Figure 2G*). An extracellular localization of the TM1-TM2 stretch was suggested by the immunofluorescent detection in non-permeabilized cells of an HA-epitope inserted at position 271 (*Figure 2E,G*). The observation that this stretch is glycosylated further supports its extracellular localization (*Figure 2F*). Only this segment displays consensus sites (N148, N155, N162, N190) for N-linked glycosylation (*Figure 2G*, *Figure 6—figure supplement 2*). Deglycosylation of WT and mutant TMEM206(Δglyc), in which all four consensus sites were eliminated, demonstrated that at least one of these sites are used (*Figure 2F*) and hence face the lumen of the ER during biogenesis. We conclude that both N- and C-termini of TMEM206 reside in the cytosol, whereas the TM1-TM2 loop faces the extracellular space (*Figure 2G*).

## Human TMEM206 mediates typical ASOR currents

Overexpression of human TMEM206 in HEK cells increased acid-activated Cl$^-$ currents about 10-fold (*Figure 3A,B*). Addition of GFP to either the amino- or carboxy-terminus of TMEM206 only moderately decreased current amplitudes compared to the untagged construct (*Figure 3B*). Currents from overexpressed hTMEM206, irrespective of whether fused to GFP, resembled native $I_{Cl,H}$. To confirm that TMEM206 is essential for the generation of $I_{Cl,H}$, we disrupted the *TMEM206* gene in both HEK and HeLa cells using CRISPR-Cas9 genomic editing. To exclude off-target effects, we used three different gRNAs to generate independent KO clones. None of the clonal HEK or HeLa KO cell lines displayed measurable $I_{Cl,H}$ currents (*Figure 3C–E*, *Figure 3—figure supplement 1*). Hence, TMEM206 is an indispensable component of ASOR.

Transfection of hTMEM206 rescued $I_{Cl,H}$ currents of *TMEM206*$^{-/-}$ (KO) HEK and HeLa cells (*Figure 3F–H*, *Figure 3—figure supplement 1E*). Current characteristics did not differ between transfected WT or KO cells (*Figure 3A,F*). *TMEM206*$^{-/-}$ HEK cells were used to further characterize hTMEM206 in transient overexpression.

Currents of overexpressed hTMEM206 (measured at acidic pH$_o$) showed the typical outward-rectification and activation kinetics of ASOR (*Capurro et al., 2015*; *Lambert and Oberwinkler, 2005*;

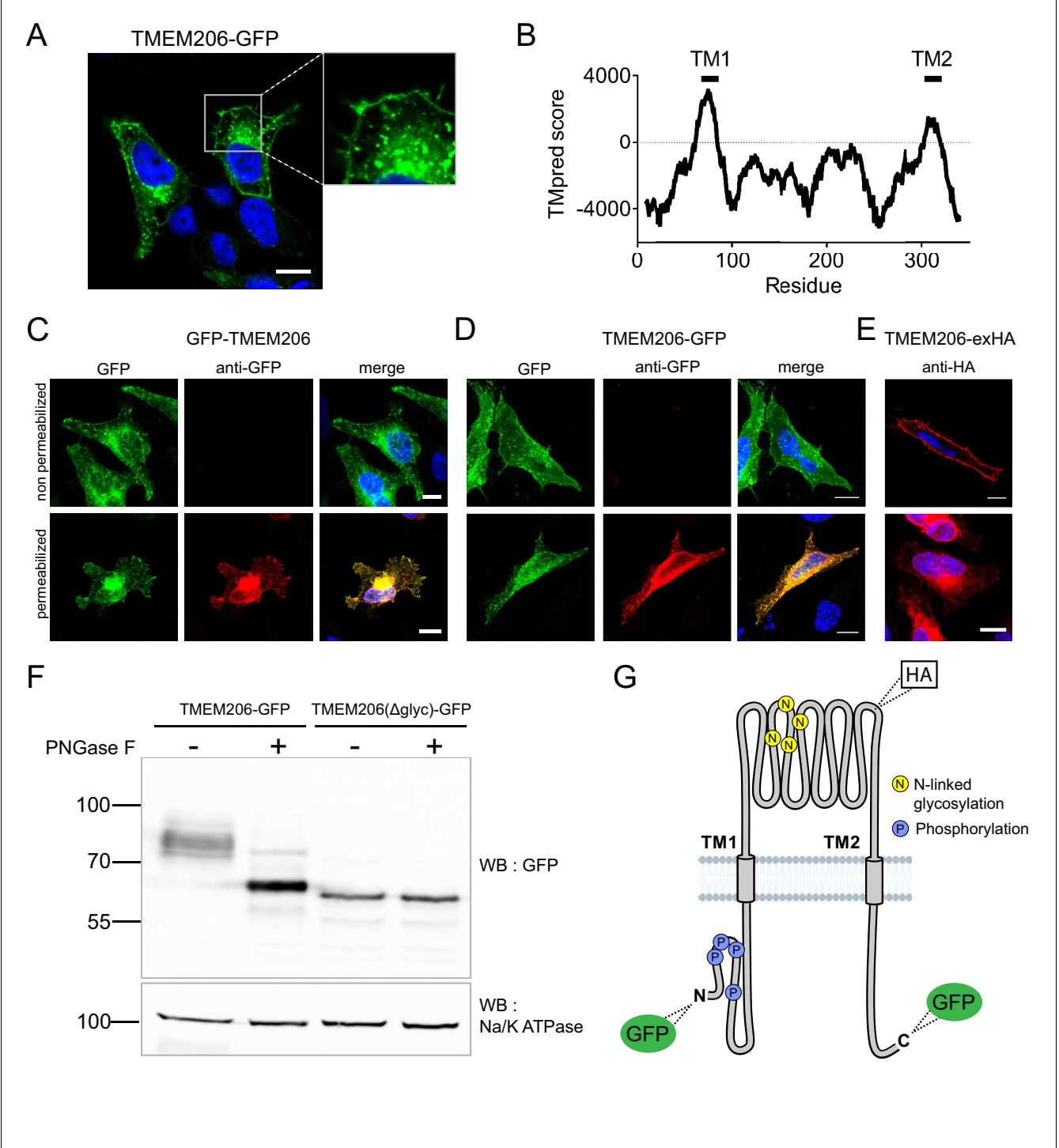

**Figure 2.** Subcellular localization and transmembrane topology of TMEM206. (A) Subcellular localization of TMEM206 (fused to GFP at the C-terminus) in transfected HeLa cells. A similar localization was observed when the tag was attached to the N-terminus. Scale bar: 10 μm (B) Hydropathy analysis of TMEM206 using the TMpred server (https://embnet.vital-it.ch/software/TMPRED_form.html) suggests the presence of two transmembrane domains. (C, D) Detection of GFP by its fluorescence (green) or immunocytochemistry (red) in cells transfected with GFP-TMEM206 (C) or TMEM206-GFP (D), without (top panels) or with (lower panels) plasma membrane permeabilization. Scale bars: 10 μm. (E) A HA-epitope inserted after residue 271 between TM1 and TM2 (G) in the TMEM206-exHA mutant was detected in non-permeabilized cells. (F) Western blot of membranes from HEK cells transfected with TMEM206-GFP or TMEM206(Δglyc)-GFP in which all four predicted N-linked glycosylation sites between TM1 and TM2 were disrupted by mutagenesis. Note the lower molecular weight of TMEM206(Δglyc)-GFP. Deglycosylation of membrane proteins by PNGaseF reduced the molecular weight of the

*Figure 2 continued on next page*

*Figure 2 continued*

WT, but not the TMEM206(Δglyc)-GFP protein. (G) Schematic topology of TMEM206. Predicted glycosylation sites (N), and phosphorylated sites (P) identified in mass spectrometry (https://www.phosphosite.org/) are indicated, as well as the location of added epitopes.

DOI: https://doi.org/10.7554/eLife.49187.005

*Wang et al., 2007*) after depolarizing voltage steps (*Figure 3C,F*). However, I/V curves for these measurements must be interpreted with some care since the very large currents (>15 nA) and the absence of series resistance compensation may have resulted in confounding voltage clamp errors. The $pH_o$-dependence of native $I_{Cl,H}$ was indistinguishable from that of TMEM206-transfected KO HEK cells (*Figure 3I,H*). The extent of acidification needed for half-activation of $I_{Cl,H}$ ($pH_o$ ~5.3) agrees with previous studies (*Capurro et al., 2015*; *Lambert and Oberwinkler, 2005*; *Sato-Numata et al., 2013*; *Wang et al., 2007*).

Measurements of reversal potentials of hTMEM206-transfected cells (*Figure 4A,B*) revealed a $SCN^->I^->NO_3^->Br^->Cl^-$ permeability sequence that is typical for ASOR (*Capurro et al., 2015*; *Lambert and Oberwinkler, 2005*). Replacement of extracellular $Cl^-$ by either gluconate$^-$ (*Figure 4C*) or $SO_4^{2-}$ (*Figure 4D*) not only abolished outward, but also inward currents which are carried by anion efflux. Since we had replaced extracellular, but not intracellular $Cl^-$, these observations suggest that gluconate$^-$ and $SO_4^{2-}$ (or $HSO_4^-$, a minor species with which $SO_4^{2-}$ is at equilibrium and whose abundance will be increased by the acidic pH) block the outward movement of $Cl^-$ through ASOR/TMEM206 channels. Alternatively, ASOR may require activation by extracellular chloride. Native $I_{Cl,H}$ is inhibited by several compounds (*Capurro et al., 2015*; *Drews et al., 2014*; *Lambert and Oberwinkler, 2005*), but none of these is specific for ASOR. Acid-activated currents from overexpressed TMEM206 were efficiently blocked by DIDS, niflumic acid, and pregnenolone sulfate (PS, *Figure 4E–H*). As described for native $I_{Cl,H}$ (*Drews et al., 2014*), the block by PS was fast and reversible (*Figure 4F*) and affected both outward and inward currents (*Figure 4G*). We conclude that overexpressed TMEM206 shares all essential properties with native ASOR currents ($I_{Cl,H}$).

## TMEM206 proteins from other species mediate moderately different $I_{Cl,H}$ currents

We asked whether TMEM206 proteins from other clades also function as ASORs and transfected orthologs from the green anole lizard (*Anolis carolinensis*), chicken (*Gallus gallus*), zebrafish (*Danio rerio*) and from the hemichordate acorn worm (*Saccoglossus kowalewski*) into $TMEM206^{-/-}$ HEK cells. We also tested TMEM206 from naked mole-rats (*Heterocephalus glaber*), mammals displaying various unusual properties including insensitivity to acidic pain (*Browe et al., 2018*; *Edrey et al., 2011*; *Park et al., 2008*) and decreased acid-induced neuronal cell death (*Husson and Smith, 2018*).

Except for the evolutionarily distant hemichordate protein, which was retained in the endoplasmic reticulum (ER), all orthologs were expressed at the plasma membrane (*Figure 5—figure supplement 1*). All plasma-membrane expressed orthologs mediated robust acid-activated outwardly rectifying $I_{Cl,H}$ currents (*Figure 5A,C*). These currents, however, differed in detail. Strikingly, TMEM206 from the reptile *Anolis carolinensis* mediated outwardly-rectifying currents already at $pH_o$ 7.4. These currents were further steeply activated when $pH_o$ dropped below 6 (*Figure 5A–D*). Whereas currents from all orthologs activated at a threshold of ~$pH_o$ 6, their currents decreased differently at $pH_o$ <5 (*Figure 5D*). The current decrease at more acidic pH questions the reliability of determining Hill coefficients for the activation of ASOR by $H^+$ ions (*Capurro et al., 2015*; *Lambert and Oberwinkler, 2005*; *Yang et al., 2019*). Differences between orthologs were also observed in depolarization-activated gating and current rectification (*Figure 5A,D*), and most importantly in anion permeability (*Figure 5E–I*). Whereas all orthologs displayed the same $Cl^-$ permeability (*Figure 5E,I*), the mammalian ASORs (human and naked mole-rat) showed a higher iodide/chloride permeability ratio than their chicken, anole and zebrafish orthologs (*Figure 5F,I*). ASORs from green anole and zebrafish displayed a significant $SO_4^{2-}$ (or $HSO_4^-$) permeability which was undetectable with the other tested orthologs (*Figure 5G–I*). These results suggest that all vertebrate TMEM206 proteins mediate ASOR currents and participate in forming its ion-selective pore.

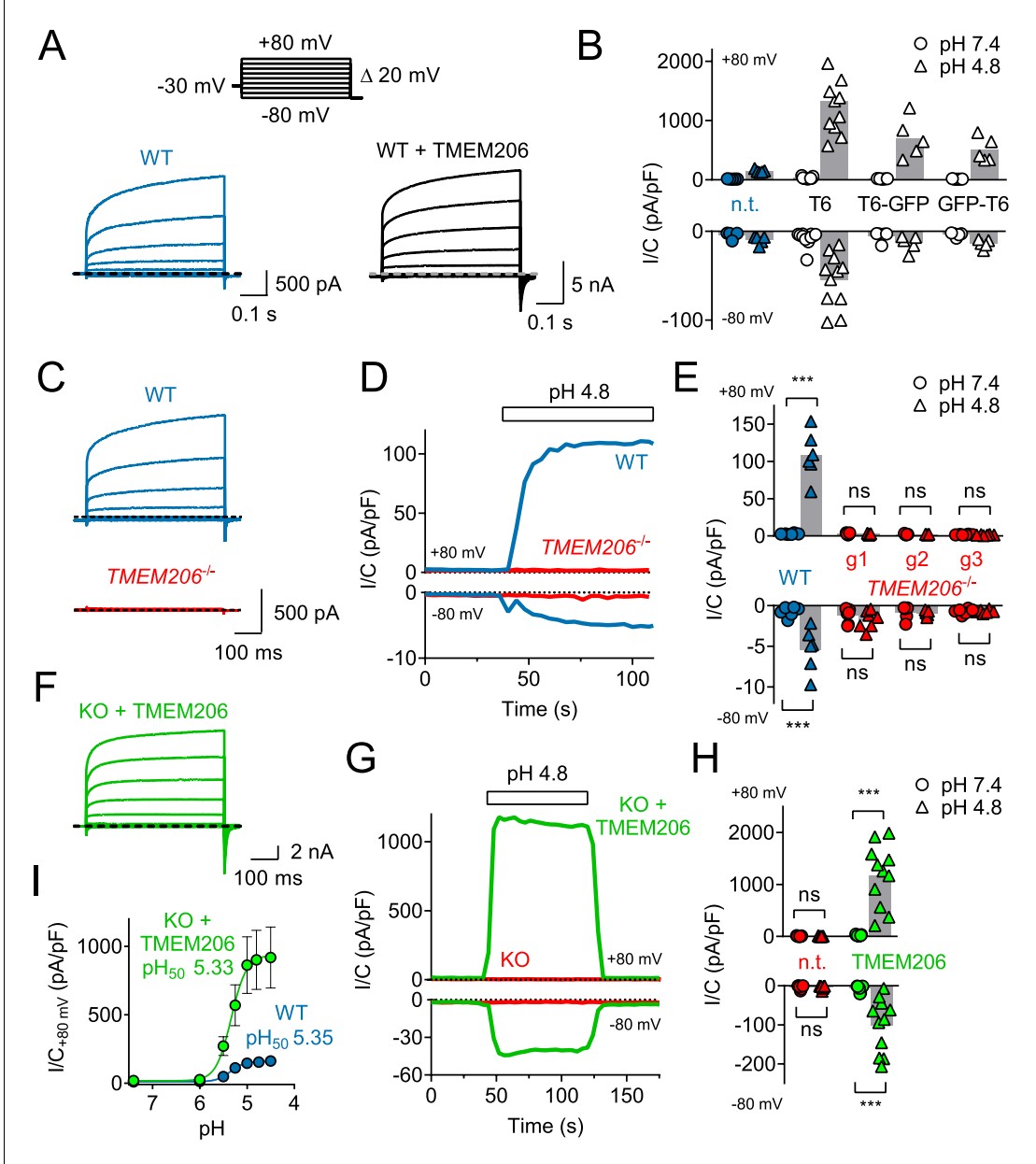

**Figure 3.** TMEM206 mediates $I_{Cl,H}$ ASOR currents. (**A**) Voltage-clamp traces of non-transfected (n.t.) and TMEM206-GFP transfected HEK cells at $pH_o = 4.8$, using the protocol shown at the top. (**B**) Current densities (I/C, at indicated $pH_o$) at +80 and −80 mV for non-transfected (n.t.) HEK cells and cells transfected with human TMEM206 (**T6**) fused N- or C-terminally to GFP, or co-expressing untagged TMEM206 with GFP from a separate pEGFP-N1 vector. Transfection increased current levels by 5- to 15-fold (bars, mean). (**C**) CRISPR-Cas9 mediated genomic disruption of TMEM206 in HEK cells (by guide RNA g1) abolished native acid-activated ASOR currents ($I_{Cl,H}$) as determined at $pH_o$ 4.8. Clamp protocol as in (**A**). (**D**) Outward and (smaller) inward $I_{Cl,H}$ currents rapidly activate in WT, but not $TMEM206^{-/-}$ HEK cells when $pH_o$ is lowered from 7.4 to 4.8. (**E**) Native $I_{Cl,H}$ of HEK cells is abolished in three independent $TMEM206^{-/-}$ cell lines generated with different guide RNAs (g1 – g3). (bars, mean; ***, p<0.001, one-way ANOVA with Bonferroni correction) (**F**) Voltage-clamp traces of $TMEM206^{-/-}$ HEK cells transfected with human TMEM206 reveal large $I_{Cl,H}$. Clamp protocol as in (**A**). (**G**) Rapid activation and deactivation of $I_{Cl,H}$ in TMEM206-transfected $TMEM206^{-/-}$ HEK cells when $pH_o$ is changed between 7.4 and 4.8. Currents monitored using a ramp protocol. (**H**) $I_{Cl,H}$ densities at indicated voltages and $pH_o$ of non-transfected (n.t.) or TMEM206-transfected $TMEM206^{-/-}$ HEK cells (bars, mean; ***, p<0.001, one-way ANOVA with Bonferroni correction). (**I**) $pH_o$-dependence of $I_{Cl,H}$ (at +80 mV) from native HEK cells and TMEM206-transfected $TMEM206^{-/-}$ HEK cells. $pH_{50}$, $pH_o$ at which current is half maximal.

DOI: https://doi.org/10.7554/eLife.49187.006

The following source data and figure supplement are available for figure 3:

**Source data 1.** Raw data for *Figure 3*.
DOI: https://doi.org/10.7554/eLife.49187.008

*Figure 3 continued on next page*

*Figure 3 continued*

**Figure supplement 1.** TMEM206 mediates $I_{Cl,H}$ ASOR currents in HeLa cells.

DOI: https://doi.org/10.7554/eLife.49187.007

## Cysteine-scanning mutagenesis reveals pore-lining residues of TMEM206

To identify pore-lining residues of ASOR we used a systematic substituted-cysteine accessibility approach (*Holmgren et al., 1996*). After having ascertained that the cysteine reagent MTSES lacks significant effects on human WT channels (*Figure 6A,B*), we generated point mutants in which all residues of both hTMEM206 transmembrane spans were singly replaced by cysteines (*Figure 6B*). Except for two mutants in TM2 (L309C and K319C), all constructs elicited acid-activated $I_{Cl,H}$ currents when transfected into *TMEM206*$^{-/-}$ HEK cells (*Figure 6B,C*, *Figure 6—figure supplement 1*). However, current amplitudes and properties were often changed (*Figure 6—figure supplement 1*). Only one mutant in TM1 (L84C) showed a marked response to cysteine modification (*Figure 6B,C*). Application of MTSES to this mutant, which changes a residue close to the predicted external end of TM1, increased currents selectively at negative voltages (*Figure 6B,C*). TM2, which is less hydrophobic than TM1 (*Figure 2B*) and may form an amphipathic α-helix (*Figure 6—figure supplement 2*), contained a larger number of responsive cysteine-substituted residues (*Figure 6B,C*). These residues were not only found close to the external end of TM2 (W304), but also close to the center (G312) and more towards the cytoplasmic end of TM2 (L315, A316) (*Figure 6B,C*). These residues failed to clearly cluster on one side of the postulated α-helix (*Figure 6—figure supplement 2A*). Depending

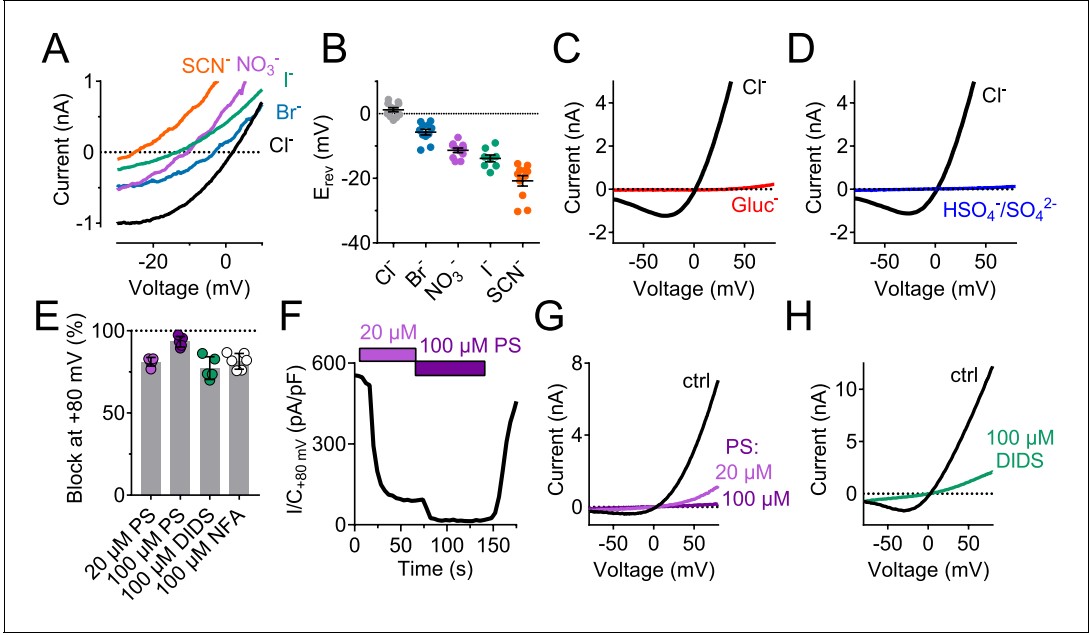

**Figure 4.** Ion selectivity and inhibitor sensitivity of TMEM206/ASOR channels. (**A**) Example traces of $I_{Cl,H}$ (elicited by voltage ramps) from TMEM206-transfected *TMEM206*$^{-/-}$ HEK cells upon equimolar replacement of extracellular NaCl (150 mM) by the sodium salts of the indicated anions. The intracellular solution contained 150 mM CsCl. The voltage at which $I = 0$ defines the reversal potential. (**B**) Reversal potentials $E_{rev}$. (**C**) Upon replacement of extracellular Cl$^-$ by gluconate, or (**D**) by HSO$_4^-$/SO$_4^{2-}$, no currents discernible from background were detected (at pH$_o$ 4.8). (**E**) Mean inhibition (%) of $I_{Cl,H}$ from TMEM206-transfected *TMEM206*$^{-/-}$ HEK cells (measured at +80 mV, pH$_o$ 4.8) by various inhibitors of ASOR. PS, pregnenolone sulfate; DIDS, 4,4'-diisothiocyano-2,2'-stilbenedisulfonic acid; NFA, niflumic acid. (**F**) Fast and reversible block of $I_{Cl,H}$ by PS. (**G**) Example I/V curves of $I_{Cl,H}$ from TMEM206-transfected *TMEM206*$^{-/-}$ HEK cells exposed to different PS concentrations. (**H**) Same for block by DIDS.

DOI: https://doi.org/10.7554/eLife.49187.010

The following source data is available for figure 4:

**Source data 1.** Raw data for *Figure 4*.

DOI: https://doi.org/10.7554/eLife.49187.011

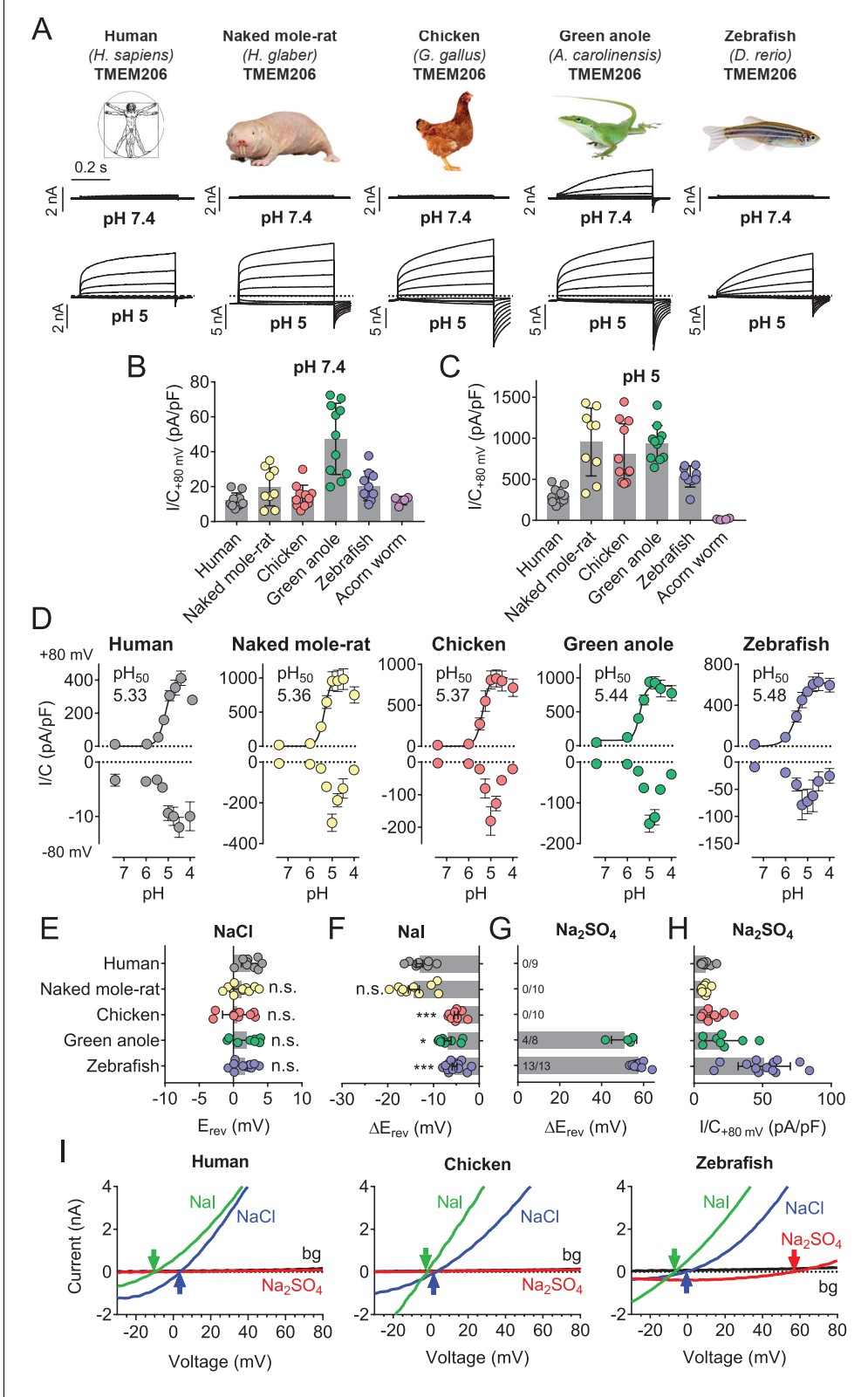

**Figure 5.** Properties of TMEM206 orthologs. (**A**) Typical voltage clamp-traces of indicated TMEM206 orthologs transiently expressed in *TMEM206*[−/−] HEK cells at $pH_o$ 7.4 (top traces) or $pH_o$ 5.0 (bottom traces). Clamp protocol as in *Figure 3A*. (**B**) Mean current amplitudes of tested orthologs at $pH_o$7.4 and (**C**) $pH_o$ 5.0. The ortholog from acorn worm failed to give currents because of its retention in the endoplasmic reticulum (*Figure 5—figure supplement 1F*). (**D**) $pH_o$-dependence of currents from various orthologs at +80 mV (top) and –80 mV (bottom) as determined from voltage ramps. (**E**–

*Figure 5 continued on next page*

*Figure 5 continued*

G) Reversal potentials $E_{rev}$ of indicated orthologs with external NaCl (E), NaI (F) and $Na_2SO_4$ (G). Significant currents with $Na_2SO_4$ could be measured only for green anole and zebrafish. $\Delta E_{rev}$, difference to $E_{rev}$ for NaCl. All currents measured at $pH_o$ 5.25. (H) Current densities (at +80 mV, $pH_o$ 5.25) with external $Na_2SO_4$ (I) Example traces showing determination of $E_{rev}$ (indicated by arrows) for human, chicken and zebrafish orthologs. bg, background current at $pH_o$ 7.4 (n = 8–13 cells; *, p<0.033; ***, p<0.001; Kruskal-Wallis test, Dunn's multiple comparison correction; error bars, SD (B,C) or SEM (D–H)).

DOI: https://doi.org/10.7554/eLife.49187.012

The following source data and figure supplement are available for figure 5:

**Source data 1.** Raw data for *Figure 5*.
DOI: https://doi.org/10.7554/eLife.49187.014

**Figure supplement 1.** Subcellular localization of different GFP-tagged TMEM206 orthologs after transfection into HeLa *TMEM206*$^{-/-}$ cells.
DOI: https://doi.org/10.7554/eLife.49187.013

on the mutant, MTSES exposure increased or decreased $I_{Cl,H}$ amplitudes (*Figure 6A–C*) or changed current rectification (*Figure 6B–C*). Consistent with a covalent modification of cysteine residues, the effects of MTSES appeared irreversible (*Figure 6A*). Strikingly, several cysteine mutants showed robust outwardly-rectifying currents already at $pH_o$ 7.4 (*Figure 6D,E*). These could be further stimulated by exposure to acidic $pH_o$ (*Figure 6F*).

Of note, some cysteine mutants displayed changed ion selectivity (*Figure 7A,B,F–H*). Whereas changes in I$^-$ permeability were relatively small (most evident for L84C and R87C, *Figure 7B,F*), the same two mutants in TM1 displayed a small sulfate permeability that was undetectable in WT hTMEM206, allowing us to determine reversal potentials (*Figure 7A,B,G–H*). Residues giving an interesting effect upon cysteine modification were further mutated to other, mostly charged, residues. This yielded detectable $I_{Cl,H}$ in many cases (*Figure 7—figure supplement 1*), with a number of mutants displaying slightly changed iodide permeability (*Figure 7C,D,F*) or marked increases in $SO_4^{2-}/HSO_4^-$ conductance (*Figure 7C,G–H*). Strikingly, replacement of L315 by negatively charged aspartate introduced a large ~22 mV shift of the reversal potential in NaCl medium (*Figure 7D,E*), indicative of a loss of anion selectivity. Together with the strong MTSES effects on the L315C mutant (*Figure 6A–C*) these data suggest that this TM2 residue importantly determines pore properties of ASOR. Of note, with the exception of R87, all functionally important residues identified here in TM1 and TM2 are highly conserved (*Figure 6—figure supplement 2B*). In conclusion, residues along the length of TM2 and towards the external end of TM1 may be close to, or line, the pore of ASOR channels.

## Role of TMEM206/ASOR channels in acidotoxicity

ASOR was suggested (*Sato-Numata et al., 2014*; *Wang et al., 2007*) to play a role in acidotoxicity because DIDS and phloretin, compounds inhibiting ASOR but also several other channels or transporters, reduced acid-induced necrotic cell death of HeLa cells (*Wang et al., 2007*) and cultured cortical neurons (*Sato-Numata et al., 2014*). We exposed both WT HEK cells, and two separately derived *TMEM206*$^{-/-}$ clones, for 2 hr to acidic medium ($pH_o$ 4.5) in the absence and presence of PS. Cell death was assessed by propidium iodide staining and normalized to the total number of cells as determined by Hoechst 33342 labeling. Indeed, disruption of TMEM206 partially protected against acid-induced cell death (*Figure 8A,B*). A roughly similar protection was seen with PS in WT but not in *TMEM206*$^{-/-}$ cells (*Figure 8A,B*), suggesting that the effect of PS on cell viability results from its ability to block ASOR (*Drews et al., 2014*). In conclusion, ASOR/TMEM206 channels enhance acidotoxicity.

This form of cell death had been attributed to cell swelling caused by osmotic gradients generated by ASOR-mediated Cl$^-$ influx (*Sato-Numata et al., 2014*; *Wang et al., 2007*). When exposed to $pH_o$ 4.2, both WT and *TMEM206*$^{-/-}$ HEK cells increased their volume within the first three minutes as indicated by increased calcein fluorescence (*Figure 8C*). Agreeing with the hypothesis (*Sato-Numata et al., 2014*; *Wang et al., 2007*) that the parallel opening of ASOR and ASICs (acid-sensitive Na$^+$-channels; *Gründer and Pusch, 2015*; *Waldmann et al., 1997*) leads to an electrically coupled influx of osmotically active Na$^+$ and Cl$^-$, WT cells swelled faster than *TMEM206*$^{-/-}$ cells. Whereas WT cells subsequently decreased their volume, *TMEM206*$^{-/-}$ cells failed to shrink (*Figure 8C*). Likewise, the ASOR inhibitor pregnenolone sulfate (100 µM) (*Drews et al., 2014*)

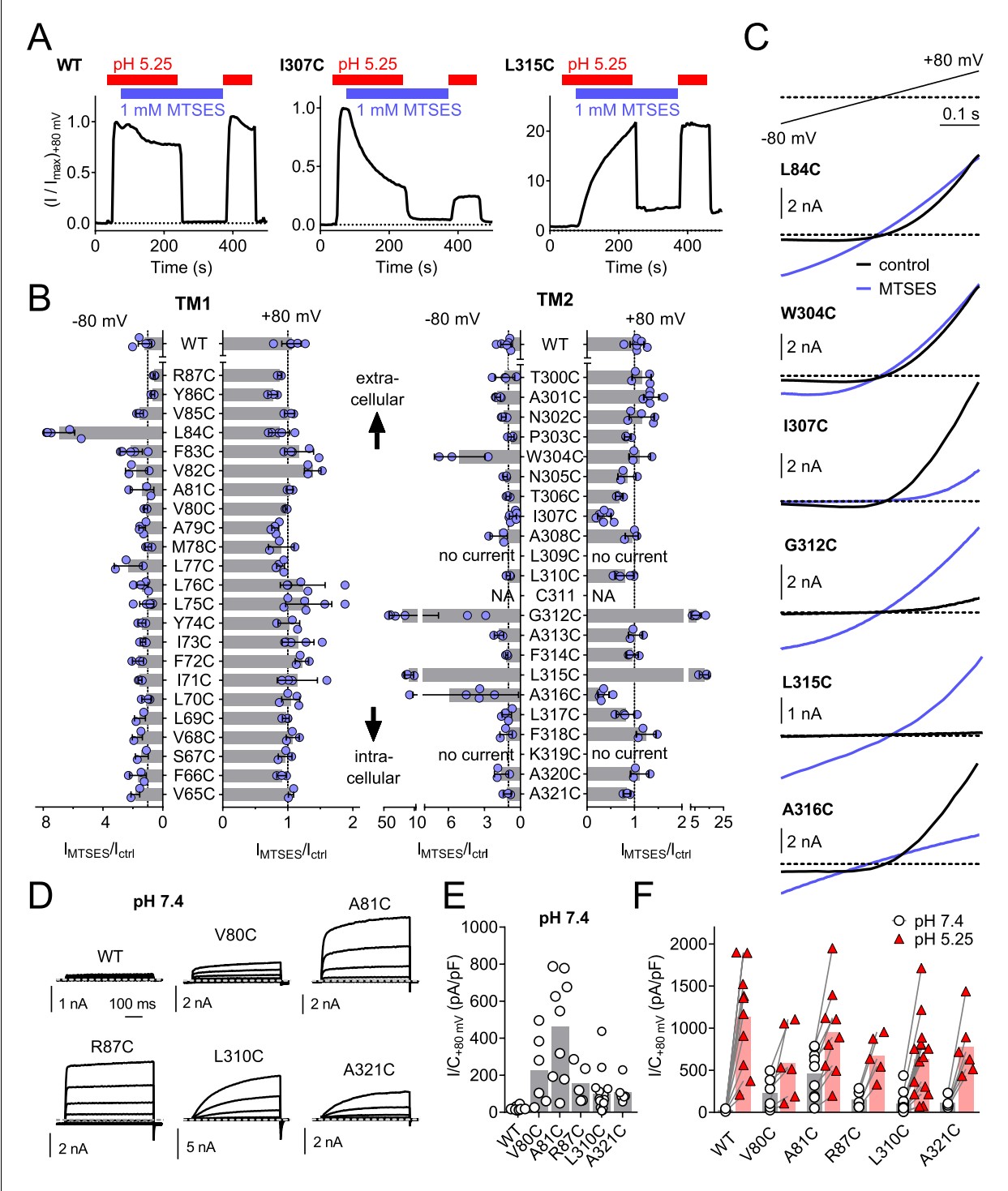

**Figure 6.** Substituted cysteine accessibility scan of TMEM206 transmembrane domains. (A) Effect of MTSES on currents from WT and mutant TMEM206-transfected *TMEM206*$^{-/-}$ cells. MTSES had no effect on WT TMEM206, but acutely and irreversibly decreased or increased $I_{Cl,H}$ of I307C and L315C mutants, respectively. Note that acidic pH$_o$ does not interfere with MTSES reactivity. The protocol was designed to be able to detect potential effects of MTSES at pH 7.4. (B) Ratio of current before ($I_{ctrl}$) and after ($I_{MTSES}$) exposure to MTSES, at +80 and −80 mV. Mutants are grouped by location within the two TMDs of TMEM206. For current amplitudes of non-modified cysteine mutants, see ***Figure 6—figure supplement 1***. (C) Current traces of selected mutants elicited by voltage ramps (top) before and after MTSES-application. (D–F) Cysteine mutants showing currents at pH$_o$ 7.4. (D) Voltage-clamp traces of WT TMEM206 and indicated mutants at pH 7.4. Clamp protocol as in ***Figure 3A***. (E) Mean current densities at pH 7.4. (F) All mutants still responded to low pH$_o$.

*Figure 6 continued on next page*

*Figure 6 continued*

DOI: https://doi.org/10.7554/eLife.49187.015

The following source data and figure supplements are available for figure 6:

**Source data 1.** Raw data for *Figure 6*.
DOI: https://doi.org/10.7554/eLife.49187.018
**Figure supplement 1.** Acid-induced current densities of cysteine mutants.
DOI: https://doi.org/10.7554/eLife.49187.016
**Figure supplement 2.** Localization of studied TMEM206 residues.
DOI: https://doi.org/10.7554/eLife.49187.017

slowed the initial swelling and inhibited the subsequent shrinkage (*Figure 8D*). Similar effects of ASOR disruption or block were observed with HeLa cells (*Figure 8—figure supplement 1*). Hence, contrasting with the conclusion of *Wang et al. (2007)*, our data suggest that the role of ASOR/ TMEM206 in acidotoxicity cannot be explained by a sustained volume increase of acid-exposed cells.

## Discussion

We have identified TMEM206 as essential, pore-forming subunit of ASOR, a widely expressed acid-sensitive outwardly rectifying anion channel. TMEM206 lacks homologous proteins within any given species. Both transmembrane domains of TMEM206, most importantly TM2, likely participate in forming its anion-selective pore. TMEM206/ASOR channels are apparently present in all vertebrates and share activation by markedly acidic pH as prominent feature. TMEM206/ASOR channels play a role in pathologies associated with a marked decrease in extracellular pH. Although their wide expression pattern across tissues and vertebrates suggests important physiological functions for all cells, these roles remain to be determined. The identification of TMEM206 as ASOR channel-forming protein in this work and in a recently published independent study by Qiu and coworkers (*Yang et al., 2019*) is an essential step forward towards the elucidation of these roles.

### TMEM206 fully constitutes ASOR

The dissimilar biophysical characteristics of currents elicited by TMEM206 orthologs and the changes observed with mutants demonstrate that TMEM206 proteins constitute the pore of ASOR channels. With only two exceptions, all cysteine-substituted channels gave currents. However, their amplitudes were reduced by several mutations which were distributed along the length of either TM1 or TM2 (*Figure 6—figure supplement 2*). Whereas this result only shows that TMEM206 is (somehow) important for ASOR currents, the observed functional effects of MTSES on various cysteine mutants additionally indicate that the respective cysteines are accessible from the aqueous phase. If such residues are located in a membrane-embedded region, they are probably close to, or line, the pore. More convincing evidence that TMEM206 proteins form the channel comes from mutations that change intrinsic channel properties such as rectification or ion selectivity (a property of the pore). Several mutations, both at the extracellular end of TM1 and at various positions along TM2, moderately changed ASOR's $I^-$ permeability. Moreover, whereas WT hTMEM206 lacked measurable currents with extracellular $Na_2SO_4$, TMEM206 from anole and zebrafish, and a surprisingly large number of mutants, displayed robust currents with extracellular sulfate. These mutations may markedly increase the channel's $SO_4^{2-}$ (or $HSO_4^-$) permeability, or relieve a channel block that was indicated by the absence of inward currents (efflux of $Cl^-$) in the presence of extracellular sulfate. The most striking evidence for TMEM206 forming ASOR's pore comes from the loss of anion selectivity with the L315D mutant. The ~22 mV positive shift of the reversal potential indicates a marked increase in cation conductance ($E_{rev}$ for $Na^+$ or $H^+$ are >+120 mV and +113 mV, respectively) that might be due to electrostatic interactions of the newly introduced negatively charged aspartate with the permeant ion. Our data suggest that the whole length of TM2, with a participation of the outer end of TM1, lines the pore of ASOR. This is reminiscent of crystallographically resolved pore structures of ASIC channels with which ASOR shares the transmembrane topology, but no obvious sequence homology. Intriguingly, ASICs and other ENaC/Deg channels display a glycine residue (in the 'GxS' motif) in TM2 that plays an important role in determining pore properties and gating

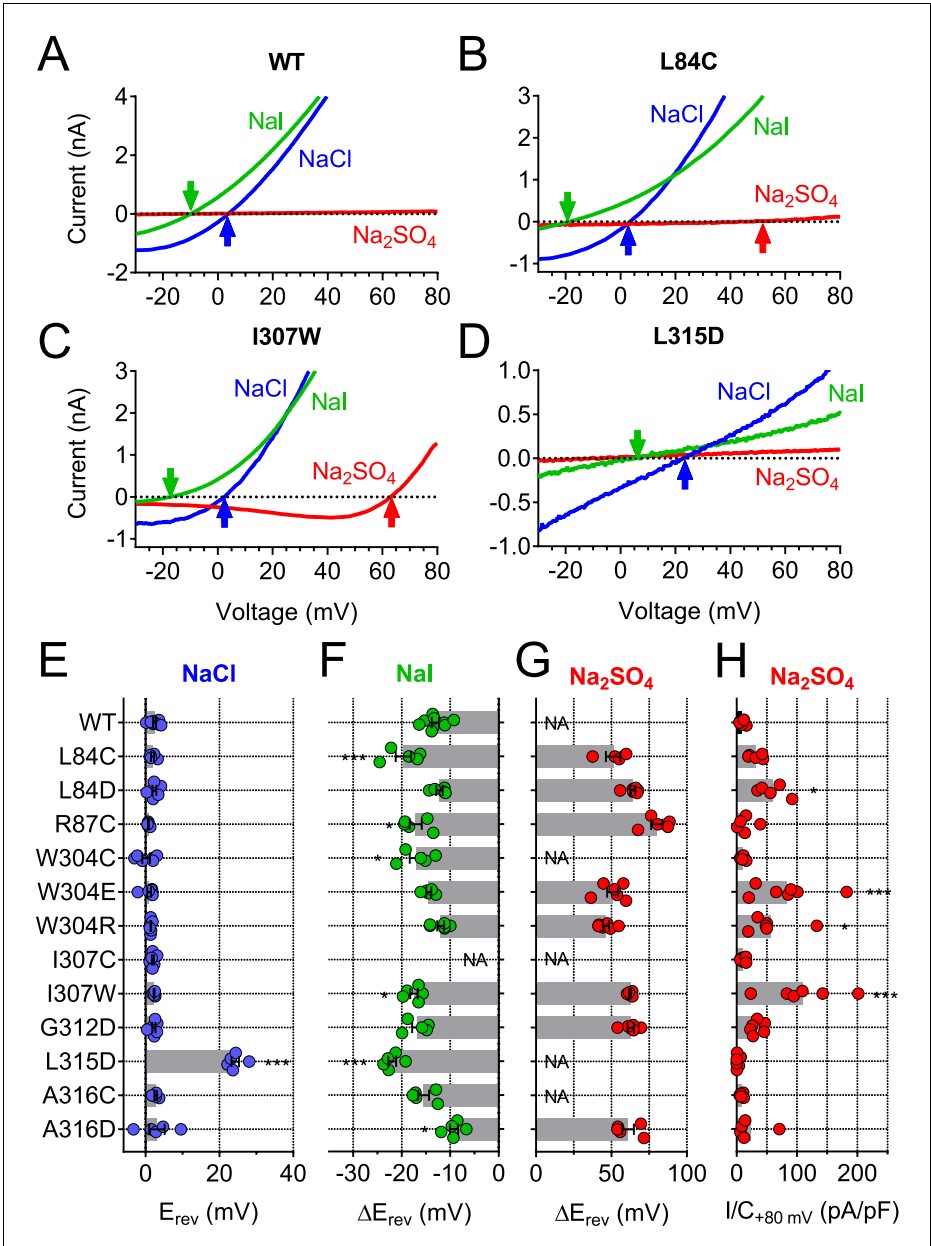

**Figure 7.** Ion selectivity changes of mutants in transmembrane domains. (**A–D**) Current traces of WT (**A**), L84C (**B**), I307W (**C**) and L315D (**D**) TMEM206 expressed in *TMEM206$^{-/-}$* HEK cells measured with indicated extracellular solutions at pH 5.25. Reversal potentials E$_{rev}$ indicated by arrows. With WT and L315D TMEM206 currents in Na$_2$SO$_4$ were too small for E$_{rev}$ determination. Currents were elicited by voltage ramps as in *Figure 6C*. (**E**) Reversal potential E$_{rev}$ with extracellular NaCl (***, p<0.001 vs. WT; one-way ANOVA, Bonferroni correction). (**F, G**) Shift of reversal potentials (ΔE$_{rev}$) with extracellular NaI (**F**) or Na$_2$SO$_4$ (**G**) relative to that measured with NaCl (NA, not applicable (currents not significantly above background or E$_{rev}$ not stable over time (I307C)). (**H**) Current densities at +80 mV with extracellular Na$_2$SO$_4$ at pH 5.25. *, p<0.033 and ***, p<0.001 vs. WT; one-way ANOVA, Bonferroni correction. No statistical analysis was performed on the data in (**G**).

DOI: https://doi.org/10.7554/eLife.49187.019

The following source data and figure supplement are available for figure 7:

**Source data 1.** Raw data for *Figure 7*.
DOI: https://doi.org/10.7554/eLife.49187.021

**Figure supplement 1.** Acid-induced current densities of TMEM206 mutants.
DOI: https://doi.org/10.7554/eLife.49187.020

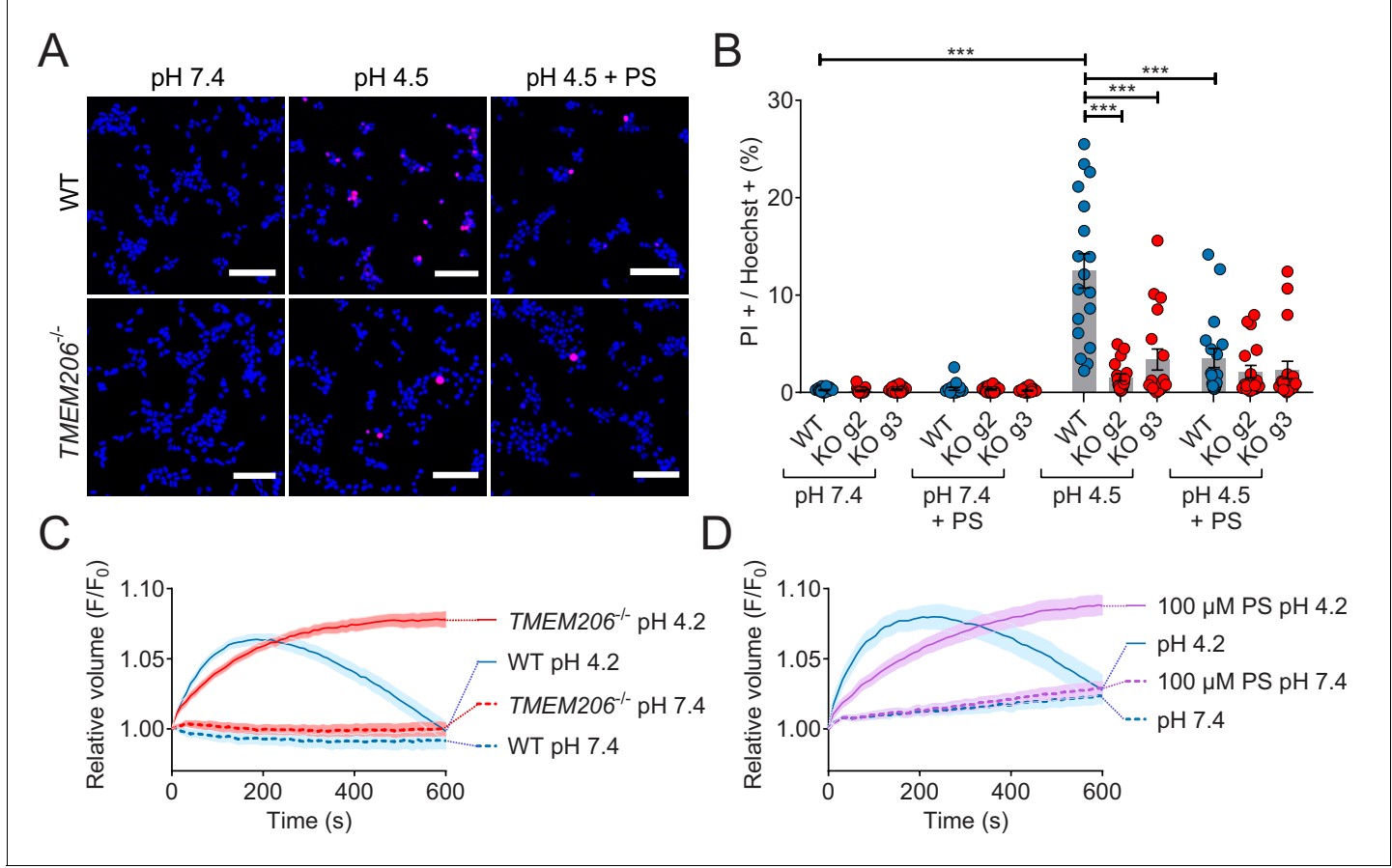

**Figure 8.** Role of TMEM206/ASOR channels in acid-induced cell death and volume regulation. (**A**) Representative pictures of WT and *TMEM206*$^{-/-}$ HEK293 cells (clone 2E5) double stained with propidium iodide (PI, red) and Hoechst 33342 (blue) after incubation for 2 hr with control (pH 7.4) or acidic (pH 4.5) solution, also containing the potent inhibitor of TMEM206 pregnenolone sulfate (PS) at 100 µM or not. In this assay, dead cells are stained with PI, membrane impermeant, whereas Hoechst 33342, which is permeant, stains nuclei of all cells. Scale bars: 100 µm (**B**) Quantification of PI positive cells over the total number of cells, determined by Hoechst staining, in WT and two independent *TMEM206*$^{-/-}$ HEK clones generated with different guide RNAs (g2, g3). For quantification, 10x objective microscopic fields were randomly chosen. Bars represent mean ± SEM. Data were acquired in three independent experiments and analyzed by using one-way ANOVA with post hoc multiple comparisons using the Tukey's test: ***p<0.001 (**C,D**) Influence of TMEM206 ablation (**C**) or block by 100 µM pregnenolone sulfate (PS) (**D**) on acid-induced cell volume changes of HEK293 cells as monitored by calcein fluorescence. Mean of eight measurements; error range, SEM. Similar results were obtained in three experiments, and similar effects were observed with HeLa cells (*Figure 8—figure supplement 1*).

DOI: https://doi.org/10.7554/eLife.49187.022

The following source data and figure supplement are available for figure 8:

**Source data 1.** Raw data for *Figure 8*.
DOI: https://doi.org/10.7554/eLife.49187.024

**Figure supplement 1.** Role of TMEM206/ASOR ablation or block on acid-induced cell volume changes in HeLa cells.
DOI: https://doi.org/10.7554/eLife.49187.023

(*Baconguis et al., 2014*; *Li et al., 2011*). Whether G212, which is located at a similar distances from the ends of TM2, plays a comparable role must await determination of TMEM206/ASOR channel structures.

Heterologous expression of all tested vertebrate orthologs led to large acid-induced currents. As shown with human TMEM206, biophysical properties of these currents did not differ significantly from native $I_{Cl,H}$. Hence completely functional ASOR channels may only require TMEM206. However, our results cannot strictly exclude that TMEM206 lines the pore together with other, unknown proteins. This scenario would require that such proteins are produced in excess endogenously since overexpression of hTMEM206 or its orthologs elicited currents > 10 fold larger than native $I_{Cl,H}$. It

further requires that such a putative additional pore-forming protein from HEK cells is able to associate with all the vertebrate orthologs tested. If ASOR's pore is exclusively constituted by TMEM206 proteins that cross the membrane only twice, TMEM206 must form at least a dimer to build a protein-enclosed pore. ASICs and other members of the DEG/ENaC channel family, which display a transmembrane topology similar to TMEM206, form trimeric homo- or heteromeric channels (*Jasti et al., 2007*). Likewise ASOR channels might be formed by TMEM206 trimers.

## Role of acid-activated chloride current

With the exception of the hemichordate TMEM206, which shows marked sequence divergence and was retained in the ER, all tested TMEM206 proteins gave strongly outwardly rectifying $Cl^-$ currents that were activated by extracellular acidification in excess of ~$pH_o$ 6. The strong rectification might be physiologically irrelevant since all orthologs gave currents also at negative voltages and because inside-positive voltages are rarely reached in vivo, in particular in non-excitable cells. By contrast, the conserved activation by acidic $pH_o$ appears crucial and poses the question where and when such pH values might be reached. Mammalian ASORs require slightly less acidic pH for activation at physiological temperatures (37°C) (*Sato-Numata et al., 2013*; *Sato-Numata et al., 2014*), but this consideration seems irrelevant for the reptile and fish orthologs.

Only a few cell types in the mammalian organism are physiologically exposed to an extracellular pH that is more acidic than pH 6.0. These include cells in the stomach and duodenum, renal medullary collecting duct (*Bengele et al., 1983*) and acid-secreting osteoclasts and macrophages (*Silver et al., 1988*), but it is unclear whether ASOR is expressed in the relevant membranes of these polarized cells. Moreover, it is unlikely that an important function in a small subset of cells has resulted in an almost ubiquitous expression of ASOR during evolution. On the other hand, intracellular compartments such as endosomes and lysosomes, which can be acidified down to pH 4.5, would provide an ideal environment for ASOR activity. Here ASOR might facilitate luminal acidification by shunting currents of the proton-ATPase as described for endosomal CLC $Cl^-$ transporters (*Günther et al., 1998*; *Jentsch, 2007*; *Piwon et al., 2000*). Interestingly, vesicular CLCs also show strong outward-rectification, but unlike ASOR are $2Cl^-/H^+$-exchangers that are inhibited by luminal (or extracellular) acidification (*Jentsch, 2007*; *Jentsch and Pusch, 2018*; *Scheel et al., 2005*). Hence presence of both ASOR and CLCs on endolysosomes might allow for a differential regulation of vesicular membrane voltage and ion concentrations. However, ClC-7 is believed to be the main $Cl^-$ transporter of lysosomes (*Graves et al., 2008*; *Kornak et al., 2001*) and immunocytochemistry showed prominent plasma membrane expression of all examined vertebrate ASOR orthologs. Although the variable cytoplasmic labeling observed upon TMEM206 overexpression may have resulted from overexpression and showed no obvious co-localization with the lysosomal marker lamp-1 in preliminary experiments, we feel that a more thorough investigation of an additional localization of ASOR in intracellular compartments may be warranted.

When speculating about biological roles of ASOR it is instructive to compare it to acid-sensitive ASIC channels (*Waldmann et al., 1997*; *Wemmie et al., 2013*). Besides being cation channels, ASICs differ from ASOR by their larger molecular diversity (four different subunits can assemble to various homo- and hetero-trimers), rapid inactivation and the degree of pH-sensitivity. Depending on the isoform, their $pH_{50}$ for activation ranges between ~4.8 and ~6.6 (*Gründer and Pusch, 2015*; *Wemmie et al., 2013*), values that are up to more than 1 pH unit more alkaline than that of ASOR ($pH_{50}$ ~5.3). Hence several physiological functions of ASICs are unlikely to have equivalents with ASOR. For instance, postsynaptic ASICs may enhance excitatory synaptic transmission by opening in response to a drop in pH in the synaptic cleft that is caused by exocytosis of the acidic contents of synaptic vesicles (*Chu and Xiong, 2012*; *Gründer and Pusch, 2015*; *Huang et al., 2015*). Even if ASOR would be expressed postsynaptically, the estimated drop in $pH_o$ (*Gründer and Pusch, 2015*) seems to be too small for its activation. ASICs also have roles in peripheral pain sensation (*Waldmann et al., 1997*; *Wemmie et al., 2013*). If expressed on appropriate sensory neurons, such a function is imaginable also for ASOR because the high intracellular $Cl^-$ concentration of peripheral neurons would allow for a depolarizing $Cl^-$-efflux (*Funk et al., 2008*). Strikingly, naked mole-rats are insensitive to acid-induced pain, but their ASIC channels show normal pH-sensitivity (*Smith et al., 2011*). We likewise found here that TMEM206/ASOR channels of that species display normal pH-sensitivity. The pH-insensitivity of mole-rats may rather be due to a changed pH-sensitivity of the $Na^+$-channel $Na_v1.7$ (*Eigenbrod et al., 2019*; *Smith et al., 2011*).

Of note, ASIC channels are involved in various neurological pathologies that are associated with acidic $pH_o$ (*Huang et al., 2015*; *Wemmie et al., 2013*), including stroke (*Xiong et al., 2004*) and multiple sclerosis (*Friese et al., 2007*). Likewise, based on protective effects of non-specific ASOR inhibitors, Okada and colleagues proposed that ASOR plays a role in acid-induced cell death of epithelial cells (*Wang et al., 2007*) and neurons (*Sato-Numata et al., 2014*). Using $TMEM206^{-/-}$ cells, we now ascertained that genetic ablation of TMEM206/ASOR channels partially protects cells from cell death provoked by strongly acidic $pH_o$ (4.5). Interestingly, when compared to mice, cortical neurons of naked mole rats are resistant to acid-induced cell death (at pH 5.0) (*Husson and Smith, 2018*). Since neither the pH-response of ASICs (*Smith et al., 2011*) nor of ASOR (this work) is changed in that species, the explanation for this resistance may be due to channel expression levels or other factors.

It has been proposed (*Wang et al., 2007*) that ASOR-dependent, acid-induced cell death results from sustained cell swelling owed to the parallel influx of $Na^+$ and $Cl^-$ through acid-activated ASIC and ASOR channels, respectively. Whereas in our experiments WT cells initially swelled faster than $TMEM206^{-/-}$ cells, they subsequently recovered their volume and even shrank. Initial swelling may indeed be owed to the opening of both ASIC and ASOR, with ASIC-mediated $Na^+$-influx mediating the depolarization required for passive $Cl^-$ influx. In contrast to ASOR, however, ASIC channels inactivate (*Gründer and Pusch, 2015*), leading to a reversal of the $Cl^-$ electrochemical potential, passive efflux of $Cl^-$ through ASOR, and subsequent osmotic cell shrinkage. Although there is now little doubt that TMEM206/ASOR channels foster acid-induced cell death, the underlying signal transduction cascade likely is complex.

Recently, while the present work was being completed, Yang et al. reported on their independent identification and characterization of TMEM206 as integral component of the ASOR channel (which the authors renamed PAC for *Proton-Activated Channel*) (*Yang et al., 2019*). They likewise reported protective effects of *TMEM206* disruption on acid-induced cell death of HEK cells or cultured primary cortical neurons. Intriguingly, these authors showed that $Tmem206^{-/-}$ mice showed smaller stroke areas in a middle cerebral artery occlusion model (*Yang et al., 2019*), although the previously reported decrease of $pH_o$ in the stroke area (down to ~6.4) (*Nedergaard et al., 1991*) is slightly above the value needed for ASOR activation. In a cysteine modification scan restricted to the extracellular half of TM2, they also found that reaction of I307C with MTSES reduces ASOR currents, and showed that the I307A mutant displayed moderately decreased iodide permeability. These results agree with our conclusion that the entire TM2 and the extracellular end of TM1 line ASOR's pore in an oligomeric complex.

## Conclusion

We have identified TMEM206 as essential subunit of the acid-sensitive outwardly rectifying anion channel ASOR. TMEM206 proteins constitute the pore of ASOR, probably in a homo-oligomeric complex. TMEM206, which lacks paralogs in any given species, defines a structurally novel class of ion channels. TMEM206/ASOR channels are involved in acid-induced cell death, but this role in pathology cannot explain its wide and possibly ubiquitous expression across tissues and vertebrate species. Its molecular identification in this work and in the parallel study of *Yang et al. (2019)* is an important step to identify its physiological roles and to mechanistically understand the diverse ways by which anion-selective channels can be formed.

## Materials and methods

**Key resources table**

| Reagent type (species) or resource | Designation | Source or reference | Identifiers | Additional information |
|---|---|---|---|---|
| Cell line (*Homo sapiens*) | HEK293 WT | Deutsche Sammlung von Mikroorganismen und Zellkulturen, Germany | ACC No. 305; RRID: CVCL_0045 | |
| Cell line (*Homo sapiens*) | HEK293 *TMEM206<sup>-/-</sup>* | this paper | | generated from WT HEK293, different clones (*Table 1*) |

*Continued on next page*

*Continued*

| Reagent type (species) or resource | Designation | Source or reference | Identifiers | Additional information |
|---|---|---|---|---|
| Cell line (*Homo sapiens*) | HeLa WT | Deutsche Sammlung von Mikroorganismen und Zellkulturen, Germany | ACC No. 57; RRID: CVCL_0030 | |
| Cell line (*Homo sapiens*) | HeLa TREx | Thermo Fisher | Cat. No. R71407; RRID: CVCL_D587 | |
| Cell line (*Homo sapiens*) | HeLa-E$^2$GFP-2A-FaNaC | this paper | see below for FaNaC/E$^2$GFP | generated from HeLa TREx, clone E2F-5 |
| Cell line (*Homo sapiens*) | HeLa *TMEM206*$^{-/-}$ | this paper | | generated from HeLa-E2GFP-2A-FaNaC, different clones (*Table 1*) |
| Transfected construct (*S. pyogenes*) | pSpCas9(BB)−2A-GFP (PX458) | PMID: 24157548 | RRID: Addgene_48138 | |
| Gene (*Homo sapiens*) | Human TMEM206 | this paper | RefSeq: NM_018252.2; UniProtKB: Q9H813 | Cloned from human brain cDNA library (Clontech) |
| Gene (*Heterocephalus glaber*) | Naked mole-rat TMEM206 | this paper | RefSeq: XM_004853543.3; UniProtKB: A0A0N8ETT8 | human codon-optimized, synthesized by Life Technologies |
| Gene (*Gallus gallus*) | Chicken TMEM206 | this paper | RefSeq: XM_419431.6; UniProtKB: A0A1D5PSZ0 | human codon-optimized, synthesized by Life Technologies |
| Gene (*Anolis carolinensis*) | Green anole TMEM206 | this paper | RefSeq: XM_003216011.3; UniProtKB: G1KFB8 | human codon-optimized, synthesized by Life Technologies |
| Gene (*Danio rerio*) | Zebrafish TMEM206 | this paper | RefSeq: NM_001291762.1; UniProtKB: X1WG42 | human codon-optimized, synthesized by Life Technologies |
| Gene (*Saccoglossus kowalevskii*) | Acorn worm TMEM206-like | this paper | RefSeq: XM_006811230.1 | human codon-optimized, synthesized by Life Technologies |
| Gene (*Cornu aspersum*) | FaNaC | PMID: 7501021; PMID: 16034422 | GenBank: X92113.1; UniProtKB: Q25011 | |
| Gene (*Aequorea victoria*) | E$^2$GFP (GFP S65T/T203Y) | PMID: 17434942, PMID: 20581829 | | subcloned from ClopHensor (Addgene #25938) |
| Antibody | anti-GFP (chicken polyclonal) | Aves Lab | Cat. No. GFP-1020; RRID: AB_10000240 | (1/1,000) |
| Antibody | anti-HA (rabbit monoclonal) | Cell Signaling | Cat. No. 3724; RRID: AB_1549585 | (1/1,000) |
| Antibody | anti-GFP (rabbit polyclonal) | Thermo Fisher | Cat. No. A-11122; RRID: AB_221569 | (1/1,000) |
| Antibody | anti-Na/K-ATPase (mouse monoclonal) | Millipore | Cat. No. 05–369; RRID: AB_309699 | (1/10,000) |
| Antibody | anti-rabbit AlexaFluor 633 (goat polyclonal) | Invitrogen | Cat. No. A21071; RRID: AB_2535732 | (1/2,000) |
| Antibody | anti-chicken HRP (rabbit polyclonal) | Sigma | Cat. No. A-9792; RRID: AB_258473 | (1/10,000) |
| Antibody | anti-mouse HRP Affinipure (goat polyclonal) | Jackson ImmunoResearch | Cat. No. 115-035-003; RRID: AB_10015289 | (1/10,000) |
| Sequence-based reagent | Dharmacon ON-TARGETplus human siRNA library | Horizon Discovery | Cat. No. G-105005-E2-025 | arrayed in 384-well plates |

*Continued on next page*

*Continued*

| Reagent type (species) or resource | Designation | Source or reference | Identifiers | Additional information |
|---|---|---|---|---|
| Chemical compound, drug | FMRFamide (H-Phe-Met-Arg -Phe-NH$_2$) | Bachem | Cat. No. 4001566 | 5 mM stock in ddH$_2$O, stored at −20°C for up to 6 months |
| Chemical compound, drug | MTSES (2-Sulfonatoethyl methanethiosulfonate) | Biotium | Cat. No. 91020 | 250 mM stock in ddH$_2$O, freshly prepared, kept on ice for up to 4 hr |
| Chemical compound, drug | Pregnenolone sulfate (PS) | Sigma | Cat. No. P162 | 50 mM stock in DMSO (stored at −20°C for up to 6 months) for electrophysiology experiments or freshly diluted in PBS for cell death analysis |
| Chemical compound, drug | 4,4′-Diisothiocya natostilbene-2,2′-disulfonic acid (DIDS) | Sigma | Cat. No. D3514 | 100 mM stock in DMSO, stored at −20°C for up to 6 months |
| Chemical compound, drug | Niflumic acid (NFA) | Sigma | Cat. No. N0630 | 300 mM stock in DMSO, stored at −20°C for up to 6 months |
| Chemical compound, drug | Hoechst 33342 | ImmunoChemistry Technologies | Cat. No. ICT-639 | |
| Chemical compound, drug | Propidium iodide | Thermo Fisher | Cat. No. P3566 | |
| Chemical compound, drug | Calcein AM | Thermo Fisher | Cat. No. 65-0853-39 | 10 mM stock in DMSO, stored at −20°C for up to 6 months |

## HeLa-E²GFP-2A-FaNaC Cell Line used in the siRNA Screen

The T-REx system (Life Technologies) was used to generate a stable HeLa cell line inducibly co-expressing the halide-sensitive E²GFP variant (GFP S65T/T203Y) (*Arosio et al., 2007*) and the FMRFamide-gated cation channel FaNaC (*Lingueglia et al., 1995*) (UniProtKB Q25011). A pcDNA5/FRT/TO-based plasmid containing the cDNAs encoding E²GFP and FaNaC, separated by a self-cleaving 2A peptide was generated for this purpose. Resulting clones were selected using 200 μg/ml hygromycin B (Santa Cruz) and 4 μg/ml blasticidin (InvivoGen). Monoclonal cell lines were tested for robust expression of E²GFP and FaNaC by fluorescence microscopy and patch clamp electrophysiology, respectively. Clone E2F-5 was chosen for the genome-wide screen. Cells were kept in DMEM supplemented with 10% tetracycline-free Hyclone FCS (Fisher Scientific) and the above-mentioned antibiotics.

## Genome-wide siRNA screen

The genome-wide siRNA screen was performed at the FMP Screening Unit using the Dharmacon ON-TARGETplus human siRNA library (Horizon Discovery) arrayed in 66 384-well plates, targeting 18090 genes by four pooled siRNAs each. The screen was performed three times.

For siRNA transfection, on a Freedom EVO 200 workstation (Tecan) 4 μl of a 500 nM library-siRNA-OptiMEM solution was mixed in each well of the 384-well assay-plate with 0.125 μl Lipofectamine RNAiMAX transfection reagent (Life Technologies) previously diluted in 5.875 μl OptiMEM (Life Technologies). Subsequently, 3,000 cells/well in phenol red-free DMEM (PAN Biotech) were seeded onto the pre-dispensed transfection mixture using an EL406TM dispenser (BioTek) resulting in a final concentration of 50 nM siRNA SMARTpool in a total volume of 40 μl per well. To induce E²GFP-2A-FaNaC expression after 24 hr, 10 μl phenol red-free DMEM containing 5 μg/ml doxycycline were added resulting in a final concentration of 1 μg/ml doxycycline.

The quenching assay was performed 72 hr post-transfection. First, using an EL406 washer dispenser (BioTek), cells were washed three times with 120 μl/well of a bath solution containing (in mM): 145 NaCl, 5 KCl, 1 MgCl$_2$, 2 CaCl$_2$, 10 glucose, 10 HEPES, pH 7.4 with NaOH, 320 mOsm/kg.

**Table 1.** TMEM206 Knockout Cell Lines.

| Cell line | Clone | sgRNA sequence | Genetic modification | Protein modification | Figures |
|---|---|---|---|---|---|
| HEK293 TMEM206⁻/⁻ | 1D4 | CAGCTGTAAGCACCATTACG | a1: duplication of a395 a2: Δa395 | 1: Y132stop between TMD1 and TMD2 2: Y132S-fs between TMD1 and TMD2 | 3E |
| | 2E5 | GGACCGAGAAGACGTTCTTC | Homozygous Δ2nt (g189-a190) | N64R-fs before TMD1 | 3 C-H; 4; 5; 6; 6-S1; 7; 7-S1; 8; |
| | 3D9 | AAGGAGACGGTCAGAGTCCA | Homozygous duplication of t110 | Q38P-fs before TMD1 | 3E; 8A-B |
| HeLa TMEM206⁻/⁻ | 1G1 | CAGCTGTAAGCACCATTACG | duplication of a395 | Y132stop between TMD1 and TMD2 | 3-S1 |
| | 2C6 | GGACCGAGAAGACGTTCTTC | a1: Δ11nt (g178-a188) a2: duplication of g189 | 1: A60E-fs 2: N64E-fs before TMD1 | 3-S1 |
| | 3F8 | AAGGAGACGGTCAGAGTCCA | duplication of t110 | Q38P-fs before TMD1 | 2A,C,D; 8C,D; 3-S1 |

a = allele; fs = frameshift; nt = nucleotide; TMD = transmembrane domain; Δ deletion.

DOI: https://doi.org/10.7554/eLife.49187.009

Subsequently, wells were aspirated to 10 µl residual volume. The plates were transferred into the FLIPR Tetra High Throughput Cellular Screening System (Molecular Devices) for measurements. All wells of the plate were simultaneously illuminated at $\lambda$ = 470–549 nm with an exposure time of 0.53 s and E²GFP-fluorescence was measured at $\lambda$ = 515–575 nm. Measurements were taken in intervals of 1 s for 300 s. After recording the baseline fluorescence for 10 s, parallel pipetting added 25 µl acidic iodide-containing solution with the following composition (in mM): 145 NaCl, 5 KCl, 1 $MgCl_2$, 2 $CaCl_2$, 10 glucose, 5 $Na_3$citrate and 30 µM FMRFamide (H-Phe-Met-Arg-Phe-$NH_2$ acetate salt, Bachem), pH 4.83, 320 mOsm/kg. The mixture of this solution with remaining bath solution resulted in approximately 100 mM iodide, 20 µM FMRFamide and pH 5. Control wells received the same solution without FMRFamide.

Analysis and processing of data obtained in the genome-wide screen were performed with the KNIME Analytics Platform (KNIME). The maximal slope of fluorescence change was determined by linear regression of 10 points in a sliding window between 20 s and 100 s. Slope was normalized to the averaged baseline and background fluorescence.

For plate-wise normalization a Z-score was calculated from the normalized slope value as follows:

$$Z = \frac{\mathrm{x}_i - \tilde{x}}{MAD_x}$$

where $\mathrm{x}_i$ is the normalized slope from well $i$, $\tilde{x}$ is the median and $MAD_x$ the median absolute deviation of all candidate wells in the plate.

The median of the three replicate Z-scores per target gene was used to rank the data. As a measure of cell viability, an additional Z-score was calculated from the baseline fluorescence. We filtered out all candidates in which the median Baseline START Z-score was lower than −0.5, indicating a significantly reduced cell number.

## Electrophysiology

Cells were plated onto poly-L-lysine-coated coverslips and transfected using the Xfect (Clontech) transfection reagent. Whole-cell patch-clamp recordings were performed 1 day after transfection using an EPC-10 USB patch-clamp amplifier and PatchMaster software (HEKA Elektronik). All experiments were performed at 20–22°C. Patch pipettes had a resistance of 1–3 MΩ in NaCl/CsCl-based solutions. The holding potential was −30 mV. Cells with a series resistance above 8 MΩ were discarded, and series resistance was not compensated for. Currents were sampled at 1 kHz and low-pass filtered at 10 kHz. In some cases, a notch filter was applied to current traces post-recording to minimize line noise.

The standard pipette solution contained (in mM): 150 CsCl, 10 EGTA, 10 HEPES, 5 $MgCl_2$, pH 7.2 with CsOH (320 mOsm/kg). The standard pH 7.4 bath solution in which cells were kept before activation of acid-induced currents contained (in mM): 145 NaCl, 5 KCl, 1 $MgCl_2$, 2 $CaCl_2$, 10 glucose,

10 HEPES, pH 7.4 with NaOH (320 mOsm/kg). Solutions with a pH between 4.0 and 5.25 were identical in composition, but buffered with 5 mM $Na_3$citrate instead of HEPES. pH of these solutions was adjusted using citrate. Likewise, solutions with a pH between 5.5 and 6.0 were buffered with 10 mM MES and adjusted with NaOH. To determine anion selectivities (*Figure 4A*), NaCl in the pH 4.8 bath solution was replaced by equimolar amounts of NaBr, NaI, $NaNO_3$ and NaSCN. The control bath solution for the ion selectivity experiments shown in *Figure 5E–I* and *Figure 7* contained (in mM) 150 NaCl, 2 $Ca(OH)_2$, 1 EGTA, five citric acid, pH 5.25 with NMDG-OH (320 mOsm/kg). To determine relative permeabilities, NaCl was replaced by an equimolar amount of NaI, or with 100 mM $Na_2SO_4$. In these experiments the pipette solution contained (in mM): 140 NMDG-Cl, 10 EGTA, 10 HEPES, 5 $MgCl_2$, pH 7.2 with NMDG-OH (320 mOsm/kg). Osmolarities of all solutions were assessed with an Osmomat 030 freezing point osmometer (Gonotec) and adjusted using mannitol.

MTSES was purchased from Biotium and stored at $-20°C$ as powder. MTSES stock solutions (250 mM in water) were freshly prepared on each day and kept on ice until final dilution in recording solutions right before experiments.

The standard protocol for measuring the time course of acid activated currents, applied every 4 s after membrane rupture, consisted of a 100 ms step to $-80$ mV followed by a 500 ms ramp from $-80$ to +80 mV. The read-out for current amplitudes was the current at $-80$ mV and +80 mV normalized to cell capacitance (current density, I/C). The voltage protocol, applied after complete activation of ASOR, consisted of 1 s steps starting from $-80$ mV to +80 mV in 20 mV intervals preceded and followed by a 0.5 s step to $-80$ mV every 4 s.

An agar bridge was used in all electrophysiological experiments. Solution was exchanged in the entire bath chamber using a custom gravity-fed perfusion system under TTL control, achieving full exchange in <5 s. Liquid junction potentials (LJPs) were only considered in ion selectivity experiments (*Figures 5* and *7*). LJPs were measured experimentally for all solutions used in these experiments with reference to NMDG-Cl pipette solution according to *Neher (1992)* and corrected for after data acquisition. The measured values (mean ± SD) were: 4.5 ± 0.3 mV (NaCl, n = 4), 4.3 ± 0.3 mV (NaI, n = 4), and 9.5 ± 0.1 mV ($Na_2SO_4$, n = 4).

## Molecular biology

Human TMEM206 was cloned from cDNA obtained from a human brain cDNA library (Clontech) using the following primers:

Forward: AGAAGCTTCCACCATGATCCGGCAGGAGCGCTCCAC
Reverse: GAGGATCCTCAGCTTATGTGGCTCGTTGCCTG

HindIII and BamHI restriction sites on the primers were used to clone the PCR product into pcDNA3.1(+) (Invitrogen). Sequencing of the entire ORF confirmed the sequence was identical to NCBI RefSeq NM_018252.2. Subsequently, the cDNA was subcloned into other vectors such as pEGFP-N1/C1 using PCR-based strategies. Point mutations were introduced using the QuikChange Kit (Agilent). All clones were verified by sequencing the entire ORF. TMEM206 ortholog cDNA clones (human codon-optimized) were synthesized by Life Technologies. The cDNAs were subcloned into pEGFP-C1 for fusion of an N-terminal GFP tag.

## Cell culture

HEK293 and HeLa cells were maintained in DMEM (PAN Biotech) supplemented with 10% FBS (PAN Biotech) and 1% penicillin/streptomycin at 37°C and 5% $CO_2$. Cells were regularly tested for mycoplasma contamination.

## Generation of monoclonal knockout cell lines

Cell lines with disruptions in the TMEM206 gene were generated using CRISPR-Cas9 as described previously (*Voss et al., 2014*). Briefly, target sgRNAs were cloned into the pSpCas9(BB)−2A-GFP (PX458, gift from Feng Zhang, MIT) vector and transfected into HEK293 and HeLa cells. 2–5 days post-transfection single GFP-positive cells were FACS-sorted into 96-well plates containing pre-conditioned DMEM medium. Monoclonal cell lines were raised and tested for sequence alterations using target-site-specific PCR on genomic DNA followed by Sanger-sequencing. sgRNA sequences, genetic modifications and resulting protein mutations are listed for each clone in *Table 1*.

## Immunocytochemistry

To determine the transmembrane topology of TMEM206, $TMEM206^{-/-}$ HeLa cells (clone 3F8) were transfected with plasmids encoding GFP-TMEM206, TMEM206-GFP or TMEM206 containing a HA tag (YPYDVPDYA) after residue 271 using Lipofectamine 2000. 24 hr after transfection, cells were fixed with 2% PFA for 10 min and permeabilized with 0.1% Triton X-100 as indicated. Cells were incubated with chicken anti-GFP (1/1000, Aves Lab) or rabbit anti-HA (1/1000, Cell Signaling) for 2 hr and subsequently with secondary antibodies coupled to AlexaFluor633 for 1 hr. For the non-permeabilized conditions, no Triton X-100 was added to the antibody solutions.

Images were acquired with a confocal microscope with a 63x NA1.4 oil-immersion lens (LSM880, Zeiss).

## Deglycosylation and western blot

To determine glycosylation of TMEM206, $TMEM206^{-/-}$ HEK cells (clone 2E5) were transfected with plasmids coding for GFP-TMEM206 or mutants of putative glycosylation sites (GFP-TMEM206 N148A, N155A, N162A, N190A) using polyethylenimine (PEI). One day after transfection, cells were scrapped in homogenization buffer containing (in mM): 140 NaCl, 20 Tris HCl pH 7.4, 2 EDTA, 4 Pefabloc (Roth) and Complete proteinase inhibitor cocktail (Roche). Cell debris were removed by centrifugation at 1000 g for 10 min and the supernatant was ultracentrifuged at 269,000 g for 30 min. Pellets containing the membrane fractions were resuspended in lysis buffer containing (in mM): 140 NaCl, 50 Tris HCl pH 6.8, 0.05 EDTA, 1% SDS, 1% Triton X-100, 2 EDTA, 4 Pefabloc (Roth) and Complete proteinase inhibitor cocktail (Roche). Protein concentration of the membrane fractions was determined by BCA.

40 µg of membrane protein were denatured by adding 2 µl of denaturation buffer (New England Biolabs), for a final volume of 20 µl, and heating the reaction at 55°C for 15 min. 4 µl of 10xG7 Buffer (NEB), 4 µl of 10% NP-40 (NEB) and 8 µl of PNGase F (NEB) were added, for a total volume of 40 µl. After 2 hr incubation at 37°C, the deglycosylation reaction was terminated by adding 10 µl of Lämmli sample buffer. Samples were separated by SDS-PAGE and analyzed by Western blot using rabbit anti-GFP antibody (1/1000, Thermo Fisher). Na/K-ATPase was used as loading control and detected using a mouse antibody (1/10,000, Millipore).

## Acid-Induced cell death assay

Acid-induced cell death of WT and $TMEM206^{-/-}$ HEK293 cells (clones 2E5 and 3D9) was assessed by double staining with Hoechst 33342 and propidium iodide (PI). WT and KO cells were incubated for 2 hr at 37°C with solution at pH 7.4 (HEPES) or 4.5 (citrate, see Electrophysiology). Experimental solutions were subsequently replaced with pH 7.4 solution containing Hoechst 33342 (1 µg/ml, ImmunoChemistry Technologies) and PI (5 µg/ml, Thermo Fischer) and cells were incubated for 15 min at 37°C. After washing, cells were fixed and analyzed using a confocal microscope (LSM880, Zeiss). For quantification, several 10x microscopic fields were randomly chosen. Percentage of PI positive cells over the total number of cells, identified with Hoechst 33342, was determined using Fiji.

## Cell volume measurements

Cell volume was measured semi-quantitatively using the calcein fluorescence method (*Capó-Aponte et al., 2005*). HEK293 cells were plated 2 days before measurements in a 96-well-plate at a density of 45,000 cells per well, HeLa cells were plated 1 day before measurements at a density of 35,000 cells per well. Cells were loaded with 10 µM calcein-AM (eBioscience) in DMEM for 1 hr at 37°C. Subsequently, cells were washed 3 times with 100 µl of a pH 7.4 (HEPES, see Electrophysiology) bath solution. After washing, each well contained 50 µl bath solution. Following a 5 min incubation period the plate was transferred into Safire$^2$ Multimode Microplate Reader (Tecan) and fluorescence measurements at λ = 495 nm were initiated. After baseline recording for 30 s, 125 µl of a pH 4.0 (citrate, see Electrophysiology) solution, or pH 7.4 bath solution were added to the wells resulting in a final pH of 4.2 or 7.4 respectively. Calcein fluorescence was measured every 10 s for 10 min. Wells containing cells of the respective cell line not loaded with calcein-AM (but otherwise treated equally) were used for background subtraction, and fluorescence values were normalized to t = 0 s (after the pipetting procedure).

## Acknowledgements

We thank Sabrina Kleissle for excellent help with the genome-wide siRNA screen, Marc Wippich for help with data analysis, and Judith von Sivers for technical assistance. This work was supported by the European Research Council (ERC) Advanced Grant VOLSIGNAL (#740537) to TJJ and an EMBO Long-Term Fellowship (ALTF 665-2017) to SB.

## Additional information

### Funding

| Funder | Grant reference number | Author |
| --- | --- | --- |
| H2020 European Research Council | Advanced Grant VOLSIGNAL (#740537) | Thomas J Jentsch |
| EMBO | Long-Term Fellowship ALTF 665-2017 | Sandy Blin |

The funders had no role in study design, data collection and interpretation, or the decision to submit the work for publication.

### Author contributions

Florian Ullrich, Conceptualization, Formal analysis, Validation, Investigation, Visualization, Writing—review and editing, Developed the genomic siRNA screening procedure, Optimized, performed and evaluated it together with KL, Performed all electrophysiology experiments; Sandy Blin, Formal analysis, Validation, Investigation, Visualization, Writing—review and editing, Performed IHC, topology, and cell death experiments; Katina Lazarow, Formal analysis, Validation, Investigation, Writing—review and editing, Together with FU evaluated the genomic siRNA screening procedure; Tony Daubitz, Formal analysis, Validation, Investigation, Visualization, Writing—review and editing, Performed cell volume measurements; Jens Peter von Kries, Resources, Supervision, Funding acquisition, Supervised siRNA screen; Thomas J Jentsch, Conceptualization, Resources, Formal analysis, Supervision, Funding acquisition, Validation, Writing—original draft, Writing—review and editing, Planned and evaluated experiments, Wrote the paper

### Author ORCIDs

Florian Ullrich  https://orcid.org/0000-0002-1153-2040
Sandy Blin  https://orcid.org/0000-0001-5762-5149
Thomas J Jentsch  https://orcid.org/0000-0002-3509-2553

### Decision letter and Author response

Decision letter https://doi.org/10.7554/eLife.49187.027
Author response https://doi.org/10.7554/eLife.49187.028

## Additional files

### Supplementary files

• Transparent reporting form
DOI: https://doi.org/10.7554/eLife.49187.025

### Data availability

Raw data are in part presented in the manuscript (e.g. IHC, Western, clamp traces), and as source data files where data points (such as current densities, ratios of permeability etc) have been extracted from original electrophysiological recordings.

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
