## [Decision Letter]

Thank you for submitting your article "Identification of TMEM206 proteins as pore of ASOR acid-sensitive chloride channels" for consideration by *eLife*. Your article has been reviewed by three peer reviewers, including Reinhard Jahn as the Reviewing Editor and Reviewer #1, and the evaluation has been overseen by Olga Boudker as the Senior Editor. The following individuals involved in review of your submission have agreed to reveal their identity: Cecilia M Canessa (Reviewer #2); Johannes Oberwinkler (Reviewer #3).

The reviewers have discussed the reviews with one another and the Reviewing Editor has drafted this decision to help you prepare a revised submission.

Summary:

Ullrich et al. present cloning and characterization of a human proton-activated chloride channel ubiquitously expressed in mammalian cells and other distant vertebrates that correspond to the protein TMEM206. This channel previously known as acid-sensitive outward rectifying chloride channel (ASOR) was also recently cloned by Yang et al. who named it proton activated chloride channel or PAC. Both studies report almost identical results with Yang also publishing a knockout mouse – which was viable and did not exhibit apparent abnormalities – but was partially protected form ischemic injury of the brain.

The work of Ullrich et al. is convincing, of high quality, and the manuscript is written in an excellent manner. Most reassuring is that both groups obtained the same findings, only differing in the response of cells exposed to low pH that first swell and later recover their volume in the presence of TMEM206.

Essential revisions:

All three reviewers were enthusiastic about the work, and all recommended acceptance of the manuscript. There are no essential revisions but some minor issues that one of the reviewers would like the authors to consider before submitting the final version.

*Reviewer #3:*

Minor points:

I personally like rather verbose Materials and methods sections, and I therefore would have wished for more information on a few occasions, specifically:

- How was the Z-score calculated, and what do the authors mean by "robust statistics"?

- How was the solution exchanged in electrophysiological experiments, and did the precise way of exchanging solutions (potentially) have an impact on liquid junction potentials? Did the authors use an agar-bridge for the reference electrode?

- How was the liquid junction potential between different solution measured? As this is not an entirely trivial procedure (with potentially different outcomes depending on the method used), it would be desirable that the authors shared the values measured, especially for those experiments where anions were replaced completely.

Normally, I would consider not employing series resistance compensation an avoidable mistake when measuring currents of several nano-amperes with a series resistance of more than 5 MΩ. In this manuscript, the choice not to use this technique has no impact on the conclusions. It may, however, have affected the shape of some of the I/V curves (for example, the linearity of the curve in Figure 4H is perhaps due to voltage-clamp errors). Maybe the authors want to point this out in an appropriate place.

The authors point out in the Introduction that the name "PAORAC" provides a better description compared to "ASOR" and in addition, it is actually the older name. This, of course, is flattering to me as I coined the term PAORAC many years back. Maybe, the authors want to consider to use the term "PAORAC" also in the title, possibly together with the newer "ASOR".

In Figure 7E, it is hard to identify that the "three stars" of the L315D mutation belong to this panel, as they are plotted between panel E and F. In Figure 7G and H, why is there no statistics reported?

---

## [Author Response]

Reviewer #3:Minor points:I personally like rather verbose Materials and methods sections, and I therefore would have wished for more information on a few occasions, specifically:- How was the Z-score calculated, and what do the authors mean by "robust statistics"?

“Robust statistics” denotes the use of median and MAD instead of mean and SD, but we agree that this is somewhat ambiguous. To clearly explain how the Z-score was calculated, we now added a formula (subsection “Genome-wide siRNA Screen”, fourth paragraph).

- How was the solution exchanged in electrophysiological experiments, and did the precise way of exchanging solutions (potentially) have an impact on liquid junction potentials? Did the authors use an agar-bridge for the reference electrode?- How was the liquid junction potential between different solution measured? As this is not an entirely trivial procedure (with potentially different outcomes depending on the method used), it would be desirable that the authors shared the values measured, especially for those experiments where anions were replaced completely.

We added more detailed descriptions of these procedures, as well as the values measured for LJPs to the Materiuals and methods section (subsection “Electrophysiology”, last paragraph).

Normally, I would consider not employing series resistance compensation an avoidable mistake when measuring currents of several nano-amperes with a series resistance of more than 5 MΩ. In this manuscript, the choice not to use this technique has no impact on the conclusions. It may, however, have affected the shape of some of the I/V curves (for example, the linearity of the curve in Figure 4H is perhaps due to voltage-clamp errors). Maybe the authors want to point this out in an appropriate place.

We thank the reviewer for pointing out this minor flaw.Series resistance in ourrecordings was generally low (2.5 – 4.5 MΩ) due to our use of patch pipettes with large openings. This should result in small voltage errors (<5 mV) when currents are in the low nA range, even without compensation. However, considering the often very high whole-cell current amplitudes in cells overexpressing TMEM206, there may indeed be a substantial voltage clamp error. We added a sentence in the Results section (subsection “Human TMEM206 mediates typical ASOR currents”, third paragraph) to caution the reader about this possibly confounding factor.

The authors point out in the Introduction that the name "PAORAC" provides a better description compared to "ASOR" and in addition, it is actually the older name. This, of course, is flattering to me as I coined the term PAORAC many years back. Maybe, the authors want to consider to use the term "PAORAC" also in the title, possibly together with the newer "ASOR".

To fairly acknowledge the earlier description of the current, we now added the PAORAC acronym (which in contrast to ASOR refers specifically to anion channels) to the title of our manuscript and to the Abstract.

In Figure 7E, it is hard to identify that the "three stars" of the L315D mutation belong to this panel, as they are plotted between panel E and F.

We changed the figure to clarify what the asterisks mean.

In Figure 7G and H, why is there no statistics reported?

We added statistics of the current densities with Na_2_SO_4_ as charge carrier in Figure 7H. However, we prefer not to perform statistical analysis on the data in Figure 7G as we believe this would mislead the reader into inferring meaningful differences between the mutants. Apparent differences are likely to be due to variable current amplitudes in Na_2_SO_4_ solutions with some mutants (see panel H), for which background currents (very negative reversal potential in our recording system) contribute more to the observed reversal potential. Also, we believe that a statistical comparison between measured reversal potentials and “not applicable” has little meaning. It is now clearly pointed out in the figure legend that no statistical analysis was performed for this panel to prevent misleading the reader.

In addition to the changes made to accommodate the suggestions of the reviewers, we also added a new figure supplement (Figure 3—figure supplement 1), which shows I_Cl,H_ measured in WT and *TMEM206*^-/-^ HeLa cells. Two additional HeLa KO clones not described in the initial submission have been measured and are now incorporated in Table 1.